# DeepGate4: Efficient and Effective Representation Learning for Circuit Design at Scale

**Ziyang Zheng**[1,5]   **Shan Huang**[2,5]   **Jianyuan Zhong**[1,5]   **Zhengyuan Shi**[1,5]   **Guohao Dai**[2,3,4]
**Ningyi Xu**[2]   **Qiang Xu**[1,5]
[1]The Chinese University of Hong Kong   [2]Shanghai Jiao Tong University   [3]Infinigence-AI
[4]Shanghai Innovation Institute   [5]National Technology Innovation Center for EDA*
{zyzheng23,jyzhong,zyzshi21,qxu}@cse.cuhk.edu.hk
{ironheart,daiguohao,xuningyi}@sjtu.edu.cn

## Abstract

Circuit representation learning has become pivotal in electronic design automation, enabling critical tasks such as testability analysis, logic reasoning, power estimation, and SAT solving. However, existing models face significant challenges in scaling to large circuits due to limitations like over-squashing in graph neural networks and the quadratic complexity of transformer-based models. To address these issues, we introduce **DeepGate4**, a scalable and efficient graph transformer specifically designed for large-scale circuits. DeepGate4 incorporates several key innovations: (1) an update strategy tailored for circuit graphs, which reduce memory complexity to sub-linear and is adaptable to any graph transformer; (2) a GAT-based sparse transformer with global and local structural encodings for AIGs; and (3) an inference acceleration CUDA kernel that fully exploit the unique sparsity patterns of AIGs. Our extensive experiments on the ITC99 and EPFL benchmarks show that DeepGate4 significantly surpasses state-of-the-art methods, achieving 15.5% and 31.1% performance improvements over the next-best models. Furthermore, the Fused-DeepGate4 variant reduces runtime by 35.1% and memory usage by 46.8%, making it highly efficient for large-scale circuit analysis. These results demonstrate the potential of DeepGate4 to handle complex EDA tasks while offering superior scalability and efficiency.

## 1 Introduction

Circuit representation learning has emerged as a crucial area in electronic design automation (EDA), reflecting the broader trend in AI of learning general representations for diverse downstream tasks (Chen et al., 2024), such as testability analysis (Shi et al., 2022), logic reasoning (Deng et al., 2024; Wu et al., 2023), power estimation (Khan et al., 2023), and SAT solving (Li et al., 2023; Shi et al., 2024a). In this domain, the DeepGate family (Li et al., 2022; Shi et al., 2023) emerges as pioneering approaches, formulating circuit netlists into graphs and utilizing graph neural networks (GNNs) to learn gate-level embeddings. DeepGate (Li et al., 2022) converts arbitrary circuit netlists into And-Inverter Graphs (AIGs) and uses logic-1 probabilities from random simulations for model supervision. Its successor, DeepGate2 (Shi et al., 2023), improves on this by learning disentangled structural and functional embeddings. In addition to the DeepGate Family, Gamora (Wu et al., 2023) extends reasoning capabilities by representing both logic gates and cones, while HOGA (Deng et al., 2024) enhances the scalability and generalizability of GNNs through hop-wise aggregation.

Despite the success on tiny circuits, inherent limitations of the GNN-based framework persist when it scales to large circuits, including difficulty in capturing long-range dependencies (Alon & Yahav, 2020), susceptibility to over-smoothing (Akansha, 2023) and over-squashing (Rusch et al., 2023), which results in poor performance on complex circuits. Consequently, DeepGate3 (Shi et al., 2024b) draws inspiration from transformer-based graph learning models by tokenizing circuits into sequences and employing graph transformers to capture global relationships within DAG-based structures. While DeepGate3 introduces fine-tuning strategies for scaling from smaller to larger

---

*This work has been partially undertaken when interning at National Technology Innovation Center for EDA.

circuits, it still struggles to handle circuits with millions of gates due to the significant memory overhead and computation redundancy of dense transformer blocks. Therefore, training an efficient and effective circuit representation learning model still remains a challenge.

In general domain, previous research on improving model efficiency has shown great potential in scaling GNNs and graph Transformers; however, significant challenges still remain in applying these advancements to circuit representation learning. These models can be broadly categorized into two types: linear graph transformers and sub-linear GNNs. On the one hand, the linear Graph Transformers, such as GraphGPS (Rampášek et al., 2022), Exphormer (Shirzad et al., 2023), Node-Former (Wu et al., 2022), DAGformer (Luo et al., 2024), and NAGphormer (Chen et al., 2022), leverage graph sparsity to perform different sparse attention, reducing memory consumption from quadratic to linear. Despite the advancement, training these models on practical circuit designs with millions or billions of gates still suffer from Out-Of-Memory(OOM) error. On the other hand, the sub-linear GNNs, such as GNNAutoScale (Fey et al., 2021), SketchGNN (Ding et al., 2022), and GraphFM (Yu et al., 2022), achieve sub-linear memory complexity by employing historical embeddings during training, with randomly sampled sub-graphs. However, these methods are primarily tailored for undirected graphs and pose challenges when applied to Directed Acyclic Graphs (DAGs). Specifically, sub-linear GNNs with random sampling strategies (Fey et al., 2021; Yu et al., 2022; Ding et al., 2022) disregard the causal relationships between sub-graphs by applying completely random sampling, resulting in suboptimal performance on function-related tasks.

In response to these challenges, we propose **DeepGate4**, an efficient and effective graph transformer specifically designed to scale to large circuits. Building on the architecture of DeepGate3 as illustrated in Figure 1, DeepGate4 utilizes GNN-based tokenizer to encode circuit function and structure. These embeddings are then processed by a transformer for global aggregation. Our approach introduces several key innovations:

- **An updating strategy** tailored for DAGs based on partitioned graph, ensuring that gate embeddings are computed in logical level order, with each gate being processed only once, thus eliminating redundant computations. While DeepGate3 is limited to fine-tuning graphs with up to 50k nodes, the proposed updating strategy, which is adaptable to any graph transformer, achieve sub-linear memory complexity and thus enable efficient training on graphs with millions of nodes.

- **A GAT-based sparse transformer** with global virtual edges, reducing both time and memory complexity to linear in a mini-batch. We further introduce structural encodings for transformers on AIGs by incorporating global and local structural encodings in initialized embedding.

- **An inference acceleration kernel**, Fused-DeepGate4, designed to optimize the inference process of tokenizer and GAT components with well-designed CUDA kernels that fully exploit the unique sparsity patterns of AIGs.

Experimental results on the ITC99 and EPFL benchmarks demonstrate that DeepGate4 significantly outperforms state-of-the-art methods, with improvements of 15.5% and 31.1% respectively over the second-best method. Furthermore, our Fused-DeepGate4 model, with inference acceleration optimizations, achieves a 41.3% reduction in runtime and 51.3% reduction in memory usage on ITC99 benchmark, and a 28.2% reduction in runtime and 32.5% reduction in memory usage on EPFL benchmark. We also evaluate the generalizability of DeepGate4 across circuits of varying scales. DeepGate4 exhibits strong generalizability, delivering outstanding performance on circuits with 400K gates, despite being trained on circuits averaging just 15K gates. Moreover, when inference on circuits ranging from 400K gates to 1.6M gates, while GNNs exhibit linear memory growth, our models maintain constant memory usage. These results suggest that DeepGate4 has potential to scale both effectively and efficiently to circuits with millions, even billions of gates.

## 2 RELATED WORK

**Circuit Representation Learning**   Circuit representation learning has become a pivotal area in electronic design automation (EDA), reflecting the broader trend in AI of learning general representations for diverse downstream tasks. In this domain, the DeepGate family (Li et al., 2022; Shi et al., 2023) emerges as pioneering approachs, exploring GNNs to encode AIGs and enabling support for a variety of EDA tasks such as testability analysis (Shi et al., 2022), power estimation (Khan et al., 2023), and SAT solving (Li et al., 2023; Shi et al., 2024a). The Gamora  (Wu et al., 2023) and HOGA  (Deng et al., 2024) further extend reasoning capabilities by representing both logic gates and cones. PolarGate (Liu et al., 2024) seeks to overcome functionality representation bottlenecks by leveraging ambipolar state principles.

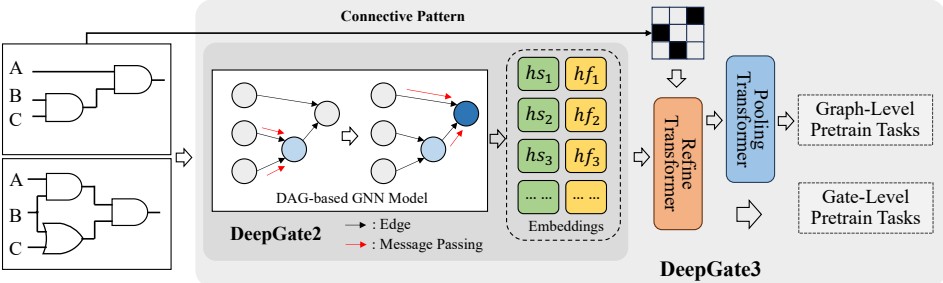

Figure 1: The overview of DeepGate2 and DeepGate3

Considering the inherent limitation of GNNs, e.g. over-squashing or over-smoothing, recent work DeepGate3 (Shi et al., 2024b), as illustrated in Figure 1, utilizes DeepGate2 as a tokenizer and then leverages the global aggregation mechanism of transformers with a connective mask to enhance circuit representation. However, new challenges arise when scaling to large AIGs: transformer-based models suffer from quadratic complexity, making training on large AIGs impractical.

**Advances and Challenges in Graph Transformers and Sub-Linear GNNs for Large-Scale Circuits**   Graph transformer models typically operate on fully-connected graphs, where every pair of nodes is connected, regardless of the original graph's structure. SAN (Kreuzer et al., 2021), Graphormer (Ying et al., 2021), GraphiT (Mialon et al., 2021), and GraphGPS (Rampášek et al., 2022) apply dense attention mechanisms with various positional and structural encodings. While these methods deliver outstanding performance, the quadratic complexity makes them impractical for large graphs. Recent approaches, such as Exphormer (Shirzad et al., 2023), Nodeformer (Wu et al., 2022), NAGformer (Chen et al., 2022), and DAGformer (Luo et al., 2024), leverage the sparse patterns of graphs to employ sparse transformers, reducing complexity to linear. However, even with these improvements, applying them to circuits with millions of gates remains challenging.

Sub-linear GNNs, such as GNNAutoScale (Fey et al., 2021) and SketchGNN (Ding et al., 2022) tackle this issue by incorporating historical embeddings during training, reducing memory complexity by reusing embeddings from prior iterations. This allows for constant GPU memory consumption relative to graph size. GraphFM (Yu et al., 2022) improves the historical embeddings updating by introducing feature momentum. However, applying them to AIGs remains challenging since they disregard the causal relationships between sub-graphs by applying completely random sampling. Specifically, when modeling circuit functionality as a computational graph, it is essential to follow a strict topological order, reasoning from primary inputs (PIs) to primary outputs (POs) based on logic levels (Li et al., 2022).

**The Necessity of System-Level GNN Optimizations for Circuit Processing**   System-level optimization of GNNs aims to reduce memory consumption and accelerate inference and training time, thereby improving the efficiency of GNN execution. Single GPU systems primarily optimize through operator reorganization, operator fusion, and data flow optimization. FuseGNN (Chen et al., 2020) accelerates the computation process by fusing any two edge-centric operators and storing intermediate data from the forward pass. However, it still consumes a large amount of memory. Fused-GAT (Zhang et al., 2022), recognized as the state-of-the-art approach, reduces redundant computations by postponing the propagation operator. It has been widely adopted in PyTorch Geometric (PyG) (PyG, 2024) implementations of the GAT network (e.g., GATConv, FuseGATConv), and its fused operators and recomputation strategy significantly reduce the memory required for execution. However, existing GNN acceleration techniques, such as Fused-GAT, were primarily designed for social network and citation network datasets, where the node degree follows a power-law distribution (Eikmeier & Gleich, 2017). In contrast, AIGs exhibit a uniform node degree distribution and have significantly fewer edges (1 or 2 edges per node). Consequently, these methods perform suboptimally on AIG graphs due to imbalanced workload and substantial synchronization overhead.

## 3    METHOD

### 3.1    OVERVIEW

The overall pipeline of our method is illustrated in Figure 2. The core idea of our method is to partition a large graph into small cones and encode these cones level by level, enabling the training

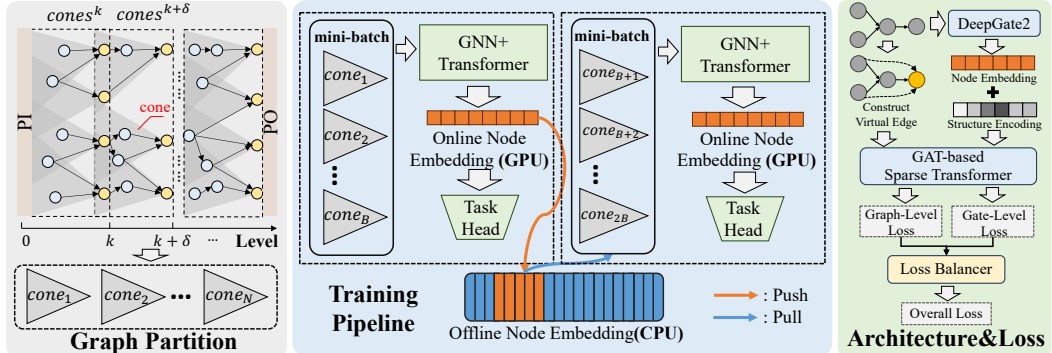

Figure 2: The overall pipeline of our method. In our training pipeline, the embedding exchanging is implemented through the following two operations: **Push(GPU to CPU)**: After encoding a mini-batch, the online node embeddings are saved in offline historical embedding. **Pull(CPU to GPU)**: Before encoding a mini-batch, the offline historical embeddings are used to initialize the online node embeddings in the overlap region.

of a graph transformer with sub-linear memory complexity. Section 3.2 details the graph partitioning process. Section 3.3 discusses our observations on overlap regions, and based on these observations, we propose the updating strategy in Section 3.4. In Section 3.5, we show the model architecture and structural encoding of our sparse transformer. Section 3.6 introduces our training objectives and a multi-task loss balancer that adjusts the weight of each component. Finally, Section 3.7 introduces inference optimization techniques to further reduce the inference runtime and memory usage.

## 3.2 GRAPH PARTITION

Given an AIG $\mathcal{G} = (V, E)$, with node set $V$, and edge set $E \subseteq V \times V$, the AIG contains three type of nodes: primary input(PI), AND gate and NOT gate. The gate type can be easily identified by its in-degree: the in-degree of a PI is 0, the in-degree of an AND gate is 2, and the in-degree of a NOT gate is 1. We first compute the logic level of each gate in topological order according to the following equation:

$$level(v) = \begin{cases} 0 & \text{if } v \text{ is a PI} \\ 1 + \max_{(u,v) \in E} level(u) & \text{otherwise} \end{cases} \tag{1}$$

For an AIG, we define a partial order $\preccurlyeq_k$ that $u \preccurlyeq_k v$ if there exists a path from $u$ to $v$ with length less than or equal to $k$. Given a node $v \in V$, based on the partial order $\preccurlyeq_k$, we define a cone by $\mathbf{cone_k}(v) = \{u \in V : u \preccurlyeq_k v\}$. Since the maximum in-degree of any node in an AIG is 2, the maximum size of $\mathbf{cone_k}(v)$ is $2^{k+1} - 1$.

As illustrated in Figure 2, given an AIG $\mathcal{G} = (V, E)$, with cone depth $k$ and stride $\delta < k$, we define the graph partition by Algorithm 1. Initially, we focus on gathering all the $cone_i^k$ that terminate at logic level $k$. Moving forward with stride $\delta$, we continue collecting with output gates situated at level $k+\delta$. Note that the chosen value of $\delta$ is smaller than $k$ in order to guarantee an overlap between cones in different level. The aforementioned process is repeated iteratively until the partitioned areas cover the entire circuit.

## 3.3 OBSERVATION AND MOTIVATION

For Intra-Level overlap, i.e. $cone_i^l \cap cone_j^l$, as shown in Figure 3a, note that if a gate $v$ is in the overlap region, then all the fan-in nodes of $v$ must be in the overlap region. Specifically, if $v \in cone_i^l \cap cone_j^l$, then $\forall u \in \{u \in cones^l : u \preccurlyeq_k v\}$, we have $u \in cone_i^l \cap cone_j^l$, since $v$ share the same fan-in region in both $cone_i^l$ and $cone_j^l$. This implies that when computing the embedding of a gate within the overlap region from scratch, the receptive field remains unchanged. When inference, since both the initialization method and model parameters are consistent, the embedding of these gates will be identical across different mini-batches.

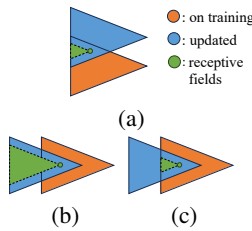

Figure 3: Observation.

For Inter-Level overlap, i.e. $cones^{l-\delta} \cap cones^l$, as illustrated in Figure 3b and Figure 3c, assume that we $v \in cones^{l-\delta} \cap cones^l$, we can define the receptive fields at different levels as follows: $R_{l-\delta}(v) = \{u \in cones^{l-\delta} : u \preceq_k v\}$ and $R_l(v) = \{u \in cones^l : u \preceq_k v\}$. According to the definition of $\preceq_k$, we observe that $R_l(v) \subseteq R_{l-\delta}(v)$, in other words, $R_l(v)$ can be regarded as $R_{l-\delta}(v)$ restricted by $cones^l$. This ensures that using historical embedding of nodes in $cones^{l-\delta}$ introduce a larger receptive filed. In contrast, computing the embedding of gate $v$ in $cones^l$ from scratch will restrict the receptive field to the current level, preventing it from capturing long-range dependencies from PIs. The receptive field affects the computations of the GNN tokenizer and sparse transformer, as they aggregate embeddings within the receptive field for node $v$. Therefore, this limitation on the receptive field will lead to significant estimation errors when performing function-related tasks (Deng et al., 2024; Liu et al., 2024).

---

**Algorithm 1** Graph Partition

---

**Input:** AIG $\mathcal{G} = (V, E)$, cone depth $k$, stride $\delta < k$
1: $L \leftarrow \max\limits_{v \in V} level(v), l \leftarrow k$
2: **while** $l \leq L$ **do**
3:    $cones^l \leftarrow list(), i \leftarrow 0$
4:    **for** $v$ in $\{v \in V : level(v) = l\}$ **do**
5:       Get sub-graph $cone_i^l \leftarrow \mathbf{cone_k}(v)$
6:       Add $cone_i^l$ to $cones^l, i \leftarrow i + 1$
7:    **end for**
8:    $l \leftarrow l + \delta$
9: **end while**
10: **for** $v$ in $\{v \in V : \text{out-degree}(v) = 0\}$ **do**
11:    Get sub-graph $g \leftarrow \mathbf{cone_k}(v)$
12:    Add $g$ to $cones^{level(v)}$
13: **end for**
14: **return** cones list $[cones^k, cones^{k+\delta}, ...]$

---

**Algorithm 2** Training Pipeline

---

**Input:**
   cone depth $k$, stride $\delta$,
   partitioned cones $[cones^k, cones^{k+\delta}, ...]$,
   mini-batch size $B$
1: **for** $l$ in $[k, k+\delta, ...]$ **do**
2:    **if** $l \neq k$ **then**
3:       Inter-Level Updating on
      $[cones^k, cones^{k+\delta}, ..., cones^l]$
4:    **end if**
5:    $m \leftarrow len(cones^l)/B$
6:    **for** $i$ in range(0,$m$) **do**
7:       sample mini-batch $batch_i^l$ in $cones^l$
8:       Intra-Level Updating on $batch_i$
9:    **end for**
10: **end for**

---

## 3.4 Updating Strategy

After partition, we get cones with level in $[k, k+\delta, ...]$. As outlined in Algorithm 2, we encode the cones starting from the smaller levels and progressing to the larger ones. Based on the observation in the Section 3.3, we propose Intra-Level Updating for cones at the same level and Inter-Level Updating for cones at different levels. Figure 4 illustrates the detailed updating process when the mini-batch size is 1.

**Intra-Level Updating** Given a cone list at the same level $cones^l = [cone_1^l, cone_2^l, ..., cone_n^l]$, we divide them into mini-batches $[batch_1^l, batch_2^l, ..., batch_m^l]$. When encoding $batch_i^l$, we first check if the gates in $batch_i^l$ have already been updated in the previous stage. If so, we retrieve their embeddings from the historical embeddings and remove all the in-edges of these gates, ensuring that their embedding will not be updated further in subsequent stages. We then send $batch_i^l$ to the model to compute the embedding of other gates, after which we will store these embedding in historical embedding and mark all the gates in $batch_i^l$ as updated in the following stage.

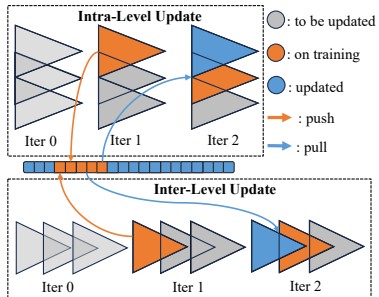

Figure 4: The updating process when the mini-batch size is 1.

**Inter-Level Updating** Given two lists of cones at different level $cones^{l-\delta}$ and $cones^l$, we ensure that $cones^{l-\delta} \cap cones^l \neq \emptyset$ due to the condition $\delta < k$ in the Algorithm 1. This mechanism allows the message from the previous level to propagate to the current level and ensures that a gate $v$ can acquire the context information from *PIs* to the current gate, i.e. a gate $v$ can aggregate information from $\{u : u \preceq_\infty v\}$, which is consistent with the information propagation flow in AIGs. The updating method is similar with Intra-Level Updating: given a cone list $cones^l$, for the gates in $cones^{l-\delta} \cap cones^l$, we will retrieve their embedding from historical embedding and remove all

the in-edges. For the updating of remaining gates, we leave them for Intra-Level Updating with $cones^l = [cone_1^l, cone_2^l, ..., cone_n^l]$.

### 3.5 GAT-BASED SPARSE TRANSFORMER

**GAT-based Sparse Attention** DAGformer (Luo et al., 2024) and DeepGate3 (Shi et al., 2024b) propose to use connective patterns as masks in transformers to effectively restrict attention in DAGs. Inspired by these approaches, we replace the Multi-head Attention module in the Transformer with a GAT module to ensure global aggregation while preserving the original transformer structure, as illustrated in Figure 5. Given a node $v \in cone_i^l$, it should aggregate information from $\{u \in cone_i^l : u \preccurlyeq_k v\}$. To achieve this, we construct virtual edges $\bar{E}$ defined as $\{(u, v) : u \preccurlyeq_k v, u \in cone_i^l\}$, which has similar function to the attention masks in DAGformer and DeepGate3. The original graph, augmented with these virtual edges, i.e. $\bar{\mathcal{G}} = (V, E \cup \bar{E})$, is then passed to the GAT-based sparse transformer to compute the embedding of each node.

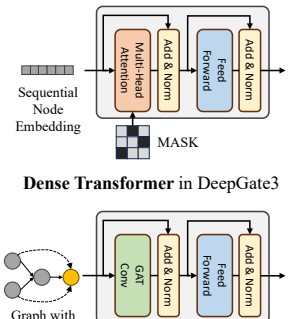

**Dense Transformer** in DeepGate3

**Sparse Transformer** in DeepGate4

Figure 5: Transformer Architecture

**Structural Encoding** In a circuit, the structure of a gate is determined by its logic level and connection pattern. Based on the aggregation mechanism of the tokenizer and sparse transformer, a gate can only acquire information from its fan-in region. However, this overlooks the out-edge pattern of a node, which is crucial for timing properties. To enhance the model's ability to capture structural information, we encode the logic level and out-degree of a gate as part of the initial structural embedding. Specifically, for a given node $v$, the structural encoding is computed by:

$$SE(v) = Emb_l(level(v)) + Emb_{and}(OutAND(v)) + Emb_{not}(OutNOT(v)), \quad (2)$$

where $Emb(\cdot)$ represents a linear layer, and $OutAND(\cdot)$ and $OutNOT(\cdot)$ denote the number of *AND* gates and *NOT* gates in $\{u : v \preccurlyeq_1 u, u \neq v\}$ respectively.

### 3.6 TRAINING OBJECTIVE

**Multi-Task Training** During the training phase of DeepGate4, we incorporate both gate-level and graph-level tasks, following the setup in DeepGate3 (Shi et al., 2024b). To separate the functional and structural embeddings, we employ training tasks with distinct labels to supervise each component:

$$L_{func} = L_{gate}^{prob} + L_{gate}^{tt\_pair} + L_{graph}^{tt} + L_{graph}^{tt\_pair} \quad (3)$$

$$L_{stru} = L_{gate}^{con} + L_{graph}^{size} + L_{graph}^{depth} + L_{graph}^{ged\_pair} + L_{in} \quad (4)$$

$$L_{all} = L_{func} + L_{stru} \quad (5)$$

For a detailed explanation of each component, please refer to Section A.2.

**Multi-Task Loss Balance** To stabilize the training process and balance the weights of each loss, inspired by previous works (Défossez et al., 2022; Chen et al., 2018), we introduce a loss balancer based on the gradient of the final layer of the sparse transformer. Given the last layer's weight $w$ and a loss $l_i$, we compute the gradient $g_i = \frac{\partial l_i}{\partial w}$. The gradient norm $\|g_i\|_2^\beta$ is computed by exponential moving average of $g_i$ with decay $\beta$. The balanced loss of $l_i$ is computed $\tilde{l}_i = \frac{l_i}{\|g_i\|_2^\beta}$ and all components are summed to form the overall loss for training.

### 3.7 INFERENCE ACCELERATION

Although the graph partitioning method provides the ability to train and infer on arbitrarily large graphs, as the number of nodes increases, the number of partitions also grows, significantly increasing the total computation time. Fused-GAT (Zhang et al., 2022) has already demonstrated excellent results by storing intermediate variables at the node level rather than the edge level, making it particularly effective for graphs with numerous edges. However, applying it directly to our tokenizer, i.e. DeepGate2, presents certain challenges: (1) the GNN component of DeepGate2 uses an aggregation mechanism similar to GAT; however, since the maximum in-degree for each node is 2, applying the Fused-GAT strategy would result in severe thread waste ((32 - 2) / 32 = 93.75%). (2)

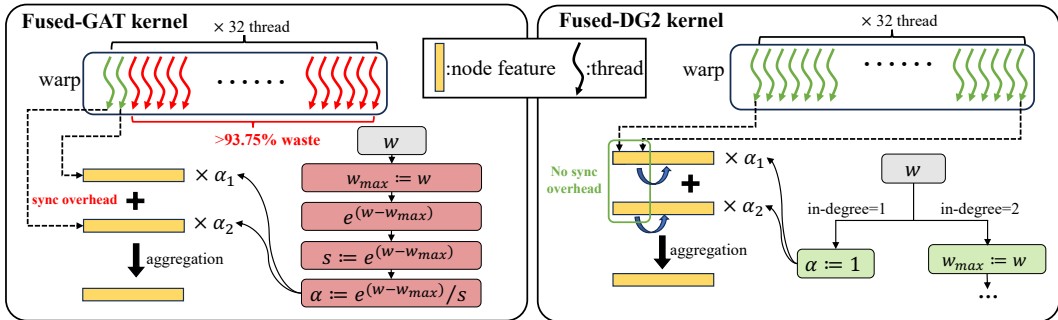

Figure 6: Comparison between Fused-GAT and our Fused-DG2 kernel. **Left**: Fused-GAT suffers from thread waste, unnecessary softmax computation when in-degree is 1, and synchronization overhead for softmax intermediate results between threads; **Right**: Fused-DG2 reallocates thread workloads, with each thread within a warp handling a portion of feature dimensions, avoiding thread waste and eliminating synchronization by independently computing attention scores, significantly reducing computation time. Furthermore, we skip the softmax computations in certain cases.

Fused-GAT calculates the softmax across many edges using the warp-level primitive *shfl_xor_sync* to synchronize the computed sum and max values, introducing substantial synchronization overhead.

**Efficient workload balance and skip computation.** We reassigned the thread computation tasks as Figure 6 shows, where each thread is responsible for the aggregation of each node, calculating all $\alpha$ values for incoming edges and performing the multiplication and accumulation, thereby avoiding the high softmax overhead, additionally, due to the characteristics of AIG graphs, where the number of edges is less than twice of nodes, storing intermediate variables at the node level is less efficient than directly storing edge information. Therefore, we switched to performing computations directly on the edges. Finally, we observed that when the in-degree is 1, we can skip the computation entirely, as the softmax result is straightforward, i.e. $\alpha = 1$. By applying these methods, we reduced both the model's inference time and memory consumption.

## 4 EXPERIMENT

### 4.1 EXPERIMENT SETTING

**Dataset** We collect the circuits from various sources, including benchmark netlists in ITC99 (Davidson, 1999) and EPFL (Amarú et al., 2015). All designs are transformed into AIGs by ABC tool (Brayton & Mishchenko, 2010). The statistical details of datasets can be found in Section A.1.

**Implementation Details** We partition the large circuits into small cones. In Algorithm 1, we set $k$ to 8 and $\delta$ to 6. The dimensions of both the structural and functional embedding are set to 128. The depth of Sparse Transformer is 12 and the depth of Pooling Transformer is 2. All training task heads are 3-layer multilayer perceptrons (MLPs). We train all models for 200 epochs to ensure convergence. The training is performed with a batch size of 1 and mini-batch size of 128 on one Nvidia A800 GPU. We utilize the Adam optimizer with a learning rate of $10^{-4}$. We report the average performance and standard deviation of the last 5 epochs, and losses without balanced weight.

### 4.2 MAIN RESULT

We compare the performance of our model with other methods on both the ITC99 and EPFL benchmarks. Table 1 presents a detailed comparison of the ITC99 benchmark across various training tasks. GNNs, such as GCN (Kipf & Welling, 2016), GraphSAGE (Hamilton et al., 2017), GAT (Veličković et al., 2017), PNA (Corso et al., 2020), DeepGate2 (Shi et al., 2023), and PolarGate (Liu et al., 2024), consume approximately 30-40 GB of GPU memory when training on ITC99, which has a maximum graph size of 140K gates. This suggests that training GNNs on circuits with more than 500K gates is impractical due to memory constraints. Sparse transformer models, such as GraphGPS (Rampášek et al., 2022), Exphormer (Shirzad et al., 2023), and DAGformer (Luo et al., 2024), also encounter OOM errors when attempting to train on ITC99, despite their linear complexity. However, with our graph partitioning and updating strategy, even dense transformer models like DeepGate3 (Shi et al., 2024b) can be successfully trained on ITC99.

Table 1: Detailed comparison experiment on ITC99 benchmark. [†]We use our graph partition and updating strategy instead of full-batch training.

| Model | Training | | Gate-level | | | | Graph-level | | | | | | | | $L_{all}$ |
|---|---|---|---|---|---|---|---|---|---|---|---|---|---|---|---|
| | Param. | Mem. | $L_{gate}^{prob}$ | $L_{gate}^{tt\text{-}pair}$ | $L_{gate}^{con}$ | $P^{con}$ | $L_{graph}^{tt}$ | $P^{tt}$ | $L_{graph}^{tt\text{-}pair}$ | $L_{graph}^{ged\text{-}pair}$ | $L_{graph}^{size}$ | $L_{graph}^{depth}$ | $L_{in}$ | $P^{in}$ | |
| GCN | 0.76M | 31.38G | 0.177 | 0.114 | 0.616 | 66.34% | 0.589 | 0.325 | 0.1596 | 0.215 | 2.65 | 1.0622 | 1.065 | 47.93% | 6.65 |
| GraphSAGE | 0.89M | 31.78G | 0.115 | 0.079 | 0.600 | 68.33% | 0.548 | 0.290 | 0.1595 | 0.203 | 2.30 | 0.9628 | 0.884 | 51.04% | 5.85 |
| GAT | 0.76M | 34.10G | 0.270 | 0.136 | 0.605 | 66.82% | 0.588 | 0.323 | 0.1601 | 0.396 | 5.32 | 0.8464 | 0.995 | 47.94% | 9.32 |
| PNA | 2.75M | 41.99G | 0.091 | 0.079 | 0.601 | 68.19% | 0.518 | 0.266 | 0.1593 | 0.181 | 3.50 | 1.0114 | 0.810 | 56.27% | 6.95 |
| GraphGPS | 6.71M | OOM | - | - | - | - | - | - | - | - | - | - | - | - | - |
| Exphormer | 0.74M | OOM | - | - | - | - | - | - | - | - | - | - | - | - | - |
| DAGformer | 1.90M | OOM | - | - | - | - | - | - | - | - | - | - | - | - | - |
| DeepGate2 | 1.28M | 32.87G | 0.049 | 0.068 | **0.594** | 68.77% | 0.513 | 0.274 | 0.1570 | 0.238 | 3.08 | 0.6772 | 0.902 | 48.62% | 6.28 |
| DeepGate3 | 8.17M | OOM | - | - | - | - | - | - | - | - | - | - | - | - | - |
| PolarGate | 0.88M | 35.95G | 0.226 | 0.100 | 0.699 | 65.92% | 0.588 | 0.326 | 0.1593 | 0.237 | 2.62 | 0.3705 | 0.688 | 52.42% | 5.69 |
| HOGA-5 | 0.78M | 42.48G | 0.204 | 0.117 | 0.609 | 68.74% | 0.493 | 0.254 | 0.1624 | 0.141 | 3.56 | 1.1378 | 0.571 | 68.99% | 6.99 |
| GraphGPS[†] | 6.71M | 7.42G | 0.109 | 0.090 | 0.632 | 66.11% | 0.434 | 0.178 | 0.1612 | 0.195 | 3.43 | 0.0061 | 0.742 | 54.62% | 5.77 |
| Exphormer[†] | 0.74M | 6.64G | 0.101 | 0.078 | 0.674 | 59.89% | 0.349 | 0.143 | 0.1160 | 0.191 | 2.32 | **0.0024** | 0.692 | 59.09% | 4.50 |
| DAGformer[†] | 1.90M | 9.52G | 0.204 | 0.116 | 0.660 | 67.53% | 0.540 | 0.243 | 0.1749 | 0.217 | 4.04 | 0.3799 | 0.705 | 57.99% | 7.01 |
| DeepGate3[†] | 8.17M | 50.75G | 0.055 | 0.061 | 0.597 | **68.93%** | 0.315 | **0.133** | **0.0780** | 0.125 | 1.93 | 0.0030 | 0.609 | 68.36% | 3.76 |
| DeepGate4 | 7.37M | 7.53G | **0.043** | **0.055** | 0.600 | 67.22% | **0.315** | 0.136 | 0.0803 | **0.117** | **1.45** | 0.0591 | **0.461** | **79.50%** | **3.16** |

**Comparison on Effectiveness** In terms of effectiveness, DeepGate4 demonstrates superior results across most training tasks. As shown in Table 3, DeepGate4 achieves state-of-the-art performance on both functional and structural tasks across the ITC99 and EPFL datasets. Regarding overall performance, DeepGate4 reduces the overall loss by 15.5% and 31.1%, respectively, compared to the second-best method. Furthermore, with the proposed structural encoding, DeepGate4 achieves a reduction of 16.4% and 34.9% in structural loss on the ITC99 and EPFL datasets, respectively.

**Comparison on Efficiency** In terms of efficiency, compared to DeepGate3[†], DeepGate4 reduces inference time and memory usage by 77.9% and 92.7% on ITC99, and by 87.8% and 95.2% on EPFL. Furthermore, with our proposed inference optimization, Fused-DeepGate4 (Fused-DG4) reduces inference time and memory usage by 41.4% and 51.4% on ITC99, and by 28.2% and 30.0% on EPFL, compared to DeepGate4.

## 4.3 Performance over Circuit of Different Scale

In this section, we discuss our model's performance across circuits of varying scales and its generalizability to Out-Of-Distribution (OOD) circuits. We trained our model on the ITC99 dataset, following the split outlined in Table 7. During training, the average graph size is 15k, while for evaluation, we used circuits of different scales, as shown in Table 2.

We extract 128 small circuits from ITC99 to ensure stable evaluation results. The average number of nodes and edges are listed as SMALL (AVG.) in Table 2. B12_OPT_C and B14_OPT_C are the original designs collected from ITC99, while MEM_CTRL is collected from EPFL. Another IMAGE_PROCESSING is the handmade design to implement multiple modes of image transformations. We employ Synopsys Design Compiler 2019.12 with skywater 130nm technology library to produce the netlist and subsequently convert it into AIG by ABC (Brayton & Mishchenko, 2010).

Table 2: Validation dataset with different scale circuits.

| name | #node | #edge | max level |
|---|---|---|---|
| small (avg.) | 161.6 | 193.9 | 46 |
| b12_opt_C | 1,861 | 2,724 | 29 |
| b14_opt_C | 10,502 | 16,135 | 96 |
| mem_ctrl | 84,742 | 130,550 | 198 |
| Image_Processing | 402,193 | 506,340 | 27 |

We present the results for circuits of varying scales in Figure 7a and Figure 7b, from which we draw three key observations. First, GNNs struggle to perform well across circuits of varying scales, while transformer-based models, such as DeepGate3[†] and DeepGate4, exhibits superior performance on both functional and structural tasks, which suggests that global aggregation mechanism is crucial in circuit representation learning. Second, with our proposed partitioning method and updating strategy, both DeepGate3[†] and DeepGate4 exhibit strong generalizability. Despite being trained on circuits averaging 15K gates, the performance on the IMAGE_PROCESSING demonstrates DeepGate3[†] and DeepGate4 maintain outstanding performance on OOD circuits. Last, DeepGate4 shows stable performance across circuits of different scales, with a standard deviation of 0.46 on overall loss. In

contrast, GNNs show unstable performance, particularly in structural loss with a standard deviation of 4.38, as highlighted in Figure 7b.

Table 3: Comparison on ITC99 and EPFL Random Control Benchmark.

| Method | ITC99 | | | | | EPFL Random Control | | | | |
|---|---|---|---|---|---|---|---|---|---|---|
| | Inference Stage | | Performance | | | Inference Stage | | Performance | | |
| | Time(s) | Mem.(MB) | $L_{func}$ | $L_{stru}$ | $L_{all}$ | Time(s) | Mem.(MB) | $L_{func}$ | $L_{stru}$ | $L_{all}$ |
| GCN | 0.297 | 415 | $1.04 \pm 0.024$ | $5.61 \pm 0.478$ | $6.65 \pm 0.471$ | 0.286 | 705 | $1.02 \pm 0.028$ | $21.16 \pm 3.023$ | $22.18 \pm 3.002$ |
| GraphSAGE | 0.020 | 415 | $0.90 \pm 0.022$ | $4.95 \pm 0.403$ | $5.85 \pm 0.391$ | 0.025 | 706 | $0.94 \pm 0.014$ | $5.94 \pm 0.923$ | $6.88 \pm 0.937$ |
| GAT | 0.029 | 415 | $1.15 \pm 0.011$ | $8.17 \pm 1.111$ | $9.32 \pm 1.102$ | 0.035 | 705 | $1.13 \pm 0.020$ | $14.46 \pm 1.335$ | $15.59 \pm 1.351$ |
| PNA | 0.042 | 423 | $0.85 \pm 0.010$ | $6.10 \pm 2.062$ | $6.95 \pm 2.065$ | 0.059 | 713 | $0.88 \pm 0.010$ | $10.10 \pm 2.218$ | $10.98 \pm 2.213$ |
| DeepGate2 | 0.490 | 412 | $0.79 \pm 0.002$ | $5.49 \pm 0.157$ | $6.28 \pm 0.158$ | 0.470 | 694 | $0.90 \pm 0.004$ | $25.78 \pm 1.546$ | $26.69 \pm 1.546$ |
| PolarGate | 0.030 | 416 | $1.07 \pm 0.014$ | $4.62 \pm 0.158$ | $5.69 \pm 0.162$ | 0.033 | 707 | $1.06 \pm 0.009$ | $9.40 \pm 2.863$ | $10.45 \pm 2.855$ |
| HOGA-5 | 0.290 | 1010 | $0.98 \pm 0.002$ | $6.02 \pm 0.290$ | $6.99 \pm 0.291$ | 0.648 | 2006 | $1.02 \pm 0.004$ | $6.33 \pm 0.290$ | $7.35 \pm 0.293$ |
| GraphGPS[†] | 0.512 | 480 | $0.78 \pm 0.020$ | $4.99 \pm 0.172$ | $5.77 \pm 0.174$ | 0.650 | 906 | $1.44 \pm 0.018$ | $11.15 \pm 0.553$ | $12.58 \pm 0.553$ |
| Exphormer[†] | 0.441 | 337 | $0.64 \pm 0.002$ | $3.86 \pm 0.207$ | $4.50 \pm 0.207$ | 0.661 | 117 | $0.85 \pm 0.027$ | $5.59 \pm 0.566$ | $6.43 \pm 0.577$ |
| DAGformer[†] | 0.676 | 1324 | $1.02 \pm 0.003$ | $5.99 \pm 0.223$ | $7.01 \pm 0.223$ | 0.886 | 292 | $1.33 \pm 0.019$ | $7.11 \pm 0.100$ | $8.43 \pm 0.091$ |
| DeepGate3[†] | 11.322 | 6565 | $0.53 \pm 0.026$ | $3.21 \pm 0.152$ | $3.73 \pm 0.148$ | 18.349 | 2730 | $1.16 \pm 0.092$ | $6.97 \pm 0.630$ | $8.13 \pm 0.697$ |
| DeepGate4 | 2.496 | 479 | $\mathbf{0.49 \pm 0.002}$ | $\mathbf{2.68 \pm 0.074}$ | $\mathbf{3.16 \pm 0.076}$ | 2.263 | 130 | $\mathbf{0.79 \pm 0.021}$ | $\mathbf{3.64 \pm 0.583}$ | $\mathbf{4.43 \pm 0.577}$ |
| Fused-DG4 | 1.463 | 233 | | | | 1.624 | 91 | | | |

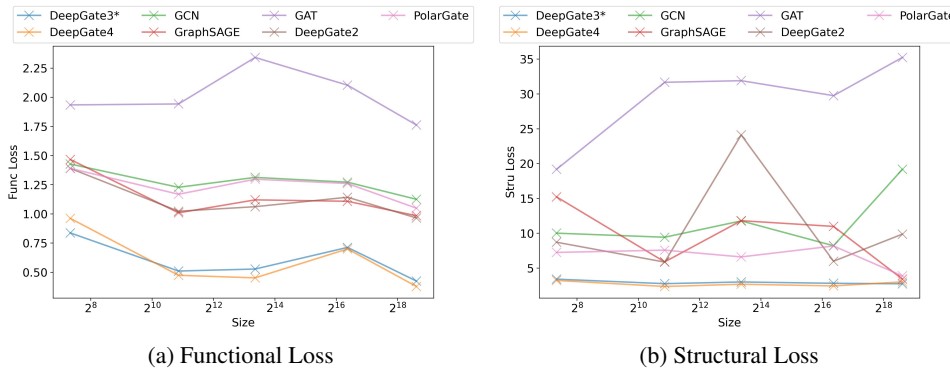

(a) Functional Loss        (b) Structural Loss

Figure 7: Performance over different scale circuits.

## 4.4 MEMORY&RUNTIME ANALYSIS

In this section, we discuss memory consumption and runtime when scaling to larger circuits. We evaluate all models on the EPFL dataset, as detailed in Table 6. For circuits larger than 400K gates, we extend the IMAGE_PROCESSING dataset in Table 2 by duplicating it 2 and 4 times to create circuits with 800K and 1.6M gates. During inference, we use a mini-batch size of 128, and drop all task heads, i.e. we use each model solely to compute embeddings.

**Inference Memory Usage** As shown in Figure 8a, for GNNs, memory usage increases linearly with graph size. For our models, memory consumption also scales linearly for small circuits. However, for circuits exceeding a certain size, the memory usage of our models stabilizes. This is primarily because, for smaller circuits, cones within the same level are smaller than the mini-batch size. These results indicate that memory consumption of our model is sub-linear with respect to graph size. Additionally, compared to DeepGate3[†], DeepGate4 demonstrates a significant reduction in memory usage, with an overall 58.4% reduction. Fused-DeepGate4 further reduces memory usage by 78.5%.

**Inference Runtime** As shown in Figure 8b, the time consumption of all models scales linearly with graph size. Since our graph transformer models partition the original graph into cones and encode them level by level, they are significantly slower than GNNs. To mitigate this, we introduce inference optimization, as described in Section 3.7. With these optimizations, our proposed Fused-DeepGate4 reduces time consumption by 90.5% and 18.5% compared to DeepGate3[†] and DeepGate4, respectively.

## 4.5 ABLATION STUDY

In this section, we perform ablation studies on the primary components of DeepGate4 following the metrics outlined in Section 3.6. All results are reported in Table 4.

Table 4: Ablation Study on ITC99 and EPFL Random Control Benchmark

| Method | ITC99 | | | | | EPFL Random Control | | | | |
| | Inference Stage | | Performance | | | Inference Stage | | Performance | | |
| | Time(s) | Mem.(MB) | $L_{func}$ | $L_{stru}$ | $L_{all}$ | Time(s) | Mem.(MB) | $L_{func}$ | $L_{stru}$ | $L_{all}$ |
|---|---|---|---|---|---|---|---|---|---|---|
| w/o Mark | 3.9223 | 1,060 | **0.48 ± 0.004** | 2.75 ± 0.041 | 3.23 ± 0.039 | 2.9906 | 182 | 0.90 ± 0.055 | 3.67 ± 0.689 | 4.58 ± 0.659 |
| w/o Partition | - | OOM | - | - | - | - | OOM | - | - | - |
| w/o Balancer&SE | 2.4591 | 479 | 0.50 ± 0.023 | 2.97 ± 0.060 | 3.47 ± 0.070 | 2.2085 | 130 | 0.82 ± 0.018 | 3.95 ± 0.847 | 4.77 ± 0.850 |
| DeepGate3[†] | 11.322 | 6565 | 0.53 ± 0.026 | 3.21 ± 0.152 | 3.73 ± 0.148 | 18.349 | 2730 | 1.16 ± 0.092 | 6.97 ± 0.630 | 8.13 ± 0.697 |
| DeepGate4 | 2.496 | 479 | 0.49 ± 0.002 | 2.68 ± 0.074 | 3.16 ± 0.076 | 2.263 | 130 | 0.79 ± 0.021 | 3.64 ± 0.583 | 4.43 ± 0.577 |
| Fused-DeepGate4 | **1.463** | **233** | | **2.68 ± 0.074** | **3.16 ± 0.076** | **1.624** | **91** | | | |

(a) Memory Consumption      (b) Time Consumption

Figure 8: Inference resource usage over different scale circuits.

**Effect of Mark** In the setting DeepGate4 without Mark (w/o Mark), not marking overlapping nodes between cones resulted in redundant computations and gradient updates. This increased average inference time and memory usage by 45.3% and 286.4%, respectively, compared to DeepGate4, demonstrating that the marking process significantly improves efficiency and reduces memory consumption without a large impact on the loss.

**Effect of Partition** Partitioning played a critical role, especially with large circuit datasets that contain a large amount of nodes. In the setting DeepGate4 without partitioning (w/o Partition), the model encounters OOM errors on both ITC99 and EPFL, highlighting the necessity of partitioning for memory usage reduction.

**Effect of Sparse Transformer** After partitioning, the inherent connectivity and sparsity in each cone allowed replacing the transformer in DeepGate3 with a sparse transformer. DeepGate4 significantly improved speed and memory efficiency, reducing inference time by 84.0% on average and reducing memory usage by 93.4% compared to the DeepGate3[†].

**Effect of Loss Balancer&Structural Encoding** The introduction of the Loss Balancer and structural encoding has almost no impact on inference time and memory usage, while significantly reducing losses, particularly the structural loss. On the two benchmarks, DeepGate4 achieved reductions of 3.38%, 8.75%, and 7.89% in functional, structural, and overall loss, respectively, compared to DeepGate4 without Loss Balancer and Structural Encoding (w/o Balancer&SE).

**Effect of Fused-DeepGate4** By introducing the SOTA GAT acceleration method, Fused-GAT, and our customized Fused-DG2 tailored specifically for the characteristics of AIGs, we further reduced both runtime and memory consumption. Compared to DeepGate4, Fused-DeepGate4 achieved an average reduction of 35.1% in inference time, and 46.8% in memory usage.

## 5 CONCLUSION

In this paper, we propose DeepGate4, an efficient and scalable representation learning model capable of handling large circuits with millions or even billions of gates. DeepGate4 introduces a novel partitioning method and update strategy applicable to any graph transformers. Additionally, it leverages a GAT-based sparse transformer with inference acceleration optimization, termed Fused-DeepGate4, specifically tailored for AIGs. Our model further incorporates global and local structural encodings, along with a loss balancer that automatically adjusts the weights of multitask losses. Experimental results on the ITC99 and EPFL benchmarks demonstrate that DeepGate4 significantly outperforms state-of-the-art methods. Moreover, the Fused-DeepGate4 variant achieves substantial reductions in both runtime and memory usage, further enhancing efficiency.

ACKNOWLEDGEMENTS

This work was supported in part by the General Research Fund of the Hong Kong Research Grants Council (RGC) under Grant No.14212422 and 14202824, and in part by National Technology Innovation Center for EDA.

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

# A APPENDIX

## A.1 DATASET STATISTIC

Table 5: OpenABC-D Dataset

| split | name | #node | #edge | #PI | #PO | max level | #cones |
|---|---|---|---|---|---|---|---|
| train | spi | 8565 | 12530 | 254 | 238 | 69 | 1348 |
| | i2c | 2195 | 3187 | 177 | 128 | 27 | 291 |
| | ss_pcm | 866 | 1165 | 104 | 90 | 13 | 144 |
| | usb_phy | 1025 | 1380 | 132 | 85 | 16 | 143 |
| | sasc | 1349 | 1827 | 135 | 124 | 15 | 165 |
| | wb_dma | 9059 | 12818 | 828 | 660 | 41 | 1779 |
| | simple_spi | 1928 | 2694 | 164 | 132 | 23 | 316 |
| | pci | 41708 | 57826 | 3429 | 3131 | 52 | 6439 |
| | dynamic_node | 36469 | 51855 | 2708 | 2560 | 55 | 5170 |
| | ac97_ctrl | 24399 | 33524 | 2339 | 2130 | 19 | 3568 |
| | mem_ctrl | 31001 | 47906 | 1187 | 937 | 56 | 3949 |
| | des3_area | 8069 | 12737 | 303 | 32 | 47 | 1226 |
| | aes | 39898 | 68140 | 683 | 529 | 44 | 4734 |
| | sha256 | 30634 | 44507 | 1943 | 1042 | 143 | 4606 |
| | fir | 9412 | 13560 | 410 | 319 | 86 | 1526 |
| | iir | 14139 | 20623 | 494 | 404 | 131 | 2377 |
| | idft | 518787 | 722736 | 37603 | 37383 | 82 | 90525 |
| | tv80 | 19877 | 30569 | 636 | 361 | 99 | 3025 |
| | fpu | 56567 | 85558 | 632 | 339 | 1522 | 8986 |
| | aes_xcrypt | 67660 | 111525 | 1975 | 1682 | 76 | 11490 |
| | jpeg | 233573 | 343382 | 4962 | 4789 | 75 | 33429 |
| | tinyRocket | 104336 | 152090 | 4561 | 4094 | 156 | 17548 |
| | picosoc | 173744 | 245387 | 11302 | 10786 | 75 | 30211 |
| | vga_lcd | 226448 | 314460 | 17322 | 17049 | 44 | 43939 |
| val | dft | 525762 | 733211 | 37597 | 37382 | 83 | 91763 |
| | wb_conmax | 83229 | 128947 | 2122 | 2032 | 35 | 12002 |
| | ethernet | 143316 | 199749 | 10731 | 10401 | 59 | 25661 |
| | bp_be | 171292 | 242214 | 11592 | 8225 | 150 | 25092 |
| | aes_secworks | 74990 | 112681 | 3087 | 2603 | 71 | 12420 |
| Avg | - | 91734.38 | 131337.5 | 5496.966 | 5160.931 | 116 | 15305.93 |

Table 6: EPFL Random Control Dataset

| split | name | #node | #edge | #PI | #PO | max level | #cones |
|---|---|---|---|---|---|---|---|
| train | router | 519 | 716 | 60 | 3 | 72 | 72 |
| | i2c | 2378 | 3584 | 136 | 127 | 36 | 311 |
| | int2float | 458 | 707 | 11 | 7 | 31 | 44 |
| | mem_ctrl | 84742 | 130550 | 1028 | 941 | 198 | 14234 |
| | voter | 27721 | 40478 | 1001 | 1 | 136 | 3822 |
| | ctrl | 328 | 495 | 7 | 25 | 19 | 64 |
| | priority | 2043 | 2893 | 128 | 8 | 498 | 262 |
| | dec | 320 | 616 | 8 | 256 | 4 | 256 |
| val | cavlc | 1298 | 1981 | 10 | 11 | 32 | 146 |
| | arbiter | 23488 | 35071 | 256 | 129 | 174 | 3714 |
| Avg | - | 14329.5 | 21709.1 | 264.5 | 150.8 | 120 | 2292.5 |

Table 7: ITC99 Dataset

| split | name | #node | #edge | #PI | #PO | max level | #cones |
|-------|------|-------|-------|-----|-----|-----------|--------|
| train | b07_opt_C | 718 | 1029 | 50 | 49 | 48 | 121 |
| | b17_opt_C | 47652 | 73743 | 1451 | 1442 | 104 | 8457 |
| | b02_opt_C | 47 | 65 | 4 | 4 | 9 | 6 |
| | b09_opt_C | 285 | 391 | 29 | 28 | 20 | 48 |
| | b05_opt_C | 956 | 1428 | 35 | 55 | 67 | 166 |
| | b15_opt_C | 14611 | 22542 | 485 | 449 | 95 | 2548 |
| | b20_opt_C | 23788 | 36709 | 522 | 508 | 102 | 4083 |
| | b13_opt_C | 538 | 720 | 62 | 53 | 23 | 75 |
| | b11_opt_C | 999 | 1487 | 38 | 31 | 56 | 134 |
| | b01_opt_C | 79 | 113 | 5 | 4 | 10 | 8 |
| | b03_opt_C | 276 | 371 | 34 | 28 | 19 | 52 |
| | b06_opt_C | 81 | 117 | 5 | 8 | 10 | 10 |
| | b04_opt_C | 1105 | 1554 | 77 | 64 | 51 | 180 |
| | b18_opt_C | 140638 | 217943 | 3306 | 3282 | 214 | 23214 |
| | b22_opt_C | 34035 | 52319 | 735 | 719 | 103 | 6133 |
| | b10_opt_C | 337 | 486 | 28 | 17 | 19 | 54 |
| | b08_opt_C | 306 | 422 | 30 | 21 | 24 | 40 |
| | b21_opt_C | 23888 | 36867 | 522 | 508 | 100 | 4403 |
| val | b12_opt_C | 1861 | 2724 | 126 | 117 | 29 | 328 |
| | b14_opt_C | 10502 | 16135 | 275 | 243 | 96 | 1751 |
| Avg | - | 15135.1 | 23358.25 | 390.95 | 381.5 | 59.95 | 2590.55 |

## A.2 TRAINING OBJECTIVE

The DeepGate4 model is trained on multiple tasks at both the gate-level and graph-level. To disentangle the functional and structural embeddings, we design training tasks with distinct labels to supervise each component.

**Gate-level Tasks.** For function-related tasks at the gate-level, we incorporate the training tasks from DeepGate2, which involve predicting the logic-1 probability of gates and the pair-wise truth table distance. We sample gate pairs, $\mathcal{N}_{gate\_tt}$, and record their corresponding simulation responses as incomplete truth tables, $T_i$. The pair-wise truth table distance $D^{gate\_tt}$ is computed as follows:

$$D_{(i,j)}^{gate\_tt} = \frac{HammingDistance(T_i, T_j)}{length(T_i)}, (i,j) \in \mathcal{N}_{gate\_tt} \tag{6}$$

The loss functions for gate-level functional tasks are:

$$L_{gate}^{prob} = L1Loss(p_k, MLP_{prob}(hf_k)), k \in \mathcal{V}$$
$$L_{gate}^{tt\_pair} = L1Loss(D_{(i,j)}^{gate\_tt}, MLP_{gate\_tt}(hf_i, hf_j)), (i,j) \in \mathcal{N}_{gate\_tt} \tag{7}$$

In addition, we incorporate supervision for structural learning by predicting pair-wise connections. Since DeepGate4 encodes the logic level as part of the structural encoding, we drop the task of predicting logic levels. The prediction of pair-wise connections is treated as a classification task, where a sampled gate pair $(i,j) \in \mathcal{N}_{gate\_con}$ can be classified into two categories: (1) there exists a path from $i$ to $j$ or from $j$ to $i$, or (2) otherwise. The loss function is defined as follows:

$$L_{gate}^{con} = BCELoss(MLP_{con}(hs_i, hs_j)), (i,j) \in \mathcal{N}_{gate\_con} \tag{8}$$

**Graph-level Tasks.** For each sub-graph, we perform a complete simulation to prepare the truth table, denoted as $T_s$. Additionally, we collect two structural characteristics for each sub-graph: the number of nodes $Size(s)$ and the depth $Depth(s)$. After obtaining the functional embedding $hf^s$ and structural embedding $hs^s$ via pooling in the Transformer, the following loss functions supervise

the training, where $s \in \mathcal{S}$:

$$
\begin{aligned}
L_{graph}^{size} &= L1Loss(Size(s), MLP_{size}(hs^s)) \\
L_{graph}^{depth} &= L1Loss(Depth(s), MLP_{depth}(hs^s)) \\
L_{graph}^{tt} &= BCELoss(T_s, MLP_{tt}(hf^s))
\end{aligned}
\tag{9}
$$

We also introduce loss functions to capture pair-wise correlations between sub-graphs. The truth table distance $D_{(s_1,s_2)}^{graph\_tt}$ and graph edit distance (Bunke, 1997) $D_{(s_1,s_2)}^{graph\_ged}$ between two sub-graphs $(s_1, s_2)$ are predicted using the following formulas:

$$
\begin{aligned}
D_{(s_1,s_2)}^{graph\_tt} &= \frac{HammingDistance(T_{s_1}, T_{s_2})}{length(T_{s_1})} \\
L_{graph}^{tt\_pair} &= L1Loss(D_{(s_1,s_2)}^{graph\_tt}, MLP_{graph\_tt}(hf^{s_1}, hf^{s_2})) \\
D_{(s_1,s_2)}^{graph\_ged} &= GraphEditDistance(s_1, s_2) \\
L_{graph}^{ged\_pair} &= L1Loss(D_{(s_1,s_2)}^{graph\_ged}, MLP_{graph\_ged}(hs^{s_1}, hs^{s_2}))
\end{aligned}
\tag{10}
$$

To link the gate-level and graph-level embeddings, we enable the model to predict whether gate $k$ belongs to sub-graph $s$ using the structural embeddings. The loss function is defined as:

$$
L_{in} = BCELoss(\{0, 1\}, MLP_{in}(hs_k, hs^s))
\tag{11}
$$

**Error of Truth Table Prediction.** For each 6-input sub-graph $s$ in the test dataset $\mathcal{S}'$, we predict the 64-bit truth table based on the graph-level functional embedding $hf^s$. The prediction error is calculated by the Hamming distance between the prediction and ground truth:

$$
P^{tt} = \frac{1}{len(\mathcal{S}')} \sum_s^{\mathcal{S}'} HammingDistance(T_s, MLP_{tt}(hf^s))
\tag{12}
$$

**Accuracy of Gate Connection Prediction.** Given the structural embedding of the gate pair $(i, j)$ in the test dataset $\mathcal{N}'_{con}$ and the binary label $y_{(i,j)}^{con} = \{0, 1\}$, we define the accuracy of gate connection prediction as:

$$
P^{con} = \frac{1}{len(\mathcal{N}'_{con})} \sum_{(i,j)}^{\mathcal{N}'_{con}} \mathbb{1}(y_{(i,j)}^{con}, MLP_{con}(hs_i, hs_j))
\tag{13}
$$

**Accuracy of Gate-in-Graph Prediction.** For each gate-graph pair $(k, s)$ in the test dataset $\mathcal{N}'_{in}$, we predict whether the gate is included in the sub-graph based on the gate structural embedding $hs_k$ and the sub-graph structural embedding $hs^s$. The binary label is $y_{(k,s)}^{in} = \{0, 1\}$. The accuracy is defined as:

$$
P^{in} = \frac{1}{len(\mathcal{N}'_{in})} \sum_{(k,s)}^{\mathcal{N}'_{in}} \mathbb{1}(MLP_{in}(hs_k, hs^s), y_k^{in})
\tag{14}
$$

### A.3 ABLATION STUDY ON GRAPH PARTITION HYPERPARAMETERS

In this section, we include a detailed analysis of the hyperparameters $k$ and $\delta$. In our graph partition algorithm, $k$ denotes the maximum level of the cone, and $\delta$ denotes the stride. These parameters influence memory usage and overlap levels as follows:

- $k$ (Maximum Level): $k$ determines the upper bound of the subgraph size. Specifically, the size of a subgraph is always smaller than $2^{k+1} - 1$. Larger subgraphs require more GPU memory; for example, with the same mini-batch size, increasing $k$ significantly increases memory consumption.
- $\delta$ (Stride): $\delta$ determines the overlap region between subgraphs. The overlap level is defined as $k - \delta + 1$, which directly influences the inter-level message-passing ratio.

Furthermore, we provide an ablation study on $k$ and $\delta$, illustrating the sensitivity of our model to these hyperparameters. As shown in Table 8, we conclude two observations from the ablation study. First, settings such as $(k = 8, \delta = 8)$, $(k = 8, \delta = 6)$, and $(k = 8, \delta = 4)$ demonstrate that our method is not sensitive to overlap ratios, as performance across these settings is similar. Second, settings such as $(k = 8, \delta = 6)$, $(k = 10, \delta = 8)$, and $(k = 6, \delta = 4)$ maintain the same overlap level but vary in subgraph size. Results demonstrate that increasing $k$ significantly impacts GPU memory usage. Furthermore, larger $k$ will degrade structural task performance. This is because structural tasks rely more heavily on local information, especially for metrics like $L_{graph}^{ged\_pair}$, $L_{graph}^{size}$, and $L_{graph}^{depth}$ (See Section A.2).

Table 8: Ablation Study on $k$ and $\delta$

| Setting | | | Metric | | |
|---|---|---|---|---|---|
| k | $\delta$ | Train Mem. | $L_{func}$ | $L_{stru}$ | $L_{all}$ |
| 8 | 8 | 12.62GB | $0.4649 \pm 0.0017$ | $2.4519 \pm 0.0625$ | $2.9168 \pm 0.0639$ |
| 8 | 6 | 12.62GB | $0.4863 \pm 0.0023$ | $2.6783 \pm 0.0739$ | $3.1646 \pm 0.0761$ |
| 8 | 4 | 12.62GB | $0.4713 \pm 0.0034$ | $2.5821 \pm 0.0963$ | $3.0534 \pm 0.0933$ |
| 10 | 8 | 33.90GB | $0.4638 \pm 0.0108$ | $3.2055 \pm 0.0747$ | $3.6692 \pm 0.0760$ |
| 6 | 4 | 6.59GB | $0.4629 \pm 0.0065$ | $2.6563 \pm 0.0587$ | $3.1192 \pm 0.0567$ |

## A.4 COMPARSION ON OPENABC-D

**Implementation Details** We collect the circuits from OpenABC-D (Chowdhury et al., 2021). All designs are transformed into AIGs by ABC tool (Brayton & Mishchenko, 2010). The statistical details of datasets can be found in Section A.1. We follow the experiment setting in Section 4.1. All experiments are performed on one L40 GPU with 48GB maximum memory. For training objectives, we use the gate-level tasks in Section A.2.

**Comparison on Effectiveness** DeepGate4 demonstrates outstanding effectiveness across all training tasks. As shown in Table 9, it achieves state-of-the-art performance on all gate-level tasks within the OpenABC-D datasets. Notably, DeepGate4 reduces the overall loss by 31.48% compared to the second-best method. Moreover, while baseline models struggle with gate connection prediction, DeepGate4 significantly enhances performance in this area, achieving an accuracy of 79%. This highlights the outstanding ability of DeepGate4 to capture the structural relationships between gates.

**Comparison on Efficiency** In terms of efficiency, models like PNA and HOGA-5 encounter out-of-memory (OOM) errors, whereas DeepGate4 can successfully train a graph transformer on large circuits containing over 500k gates.

Table 9: Comparsion on OpenABC-D benchmark.

| Model | Param. | Mem. | $L_{gate}^{prob}$ | $L_{gate}^{tt\_pair}$ | $L_{gate}^{con}$ | $P^{con}$ | $L_{all}$ |
|---|---|---|---|---|---|---|---|
| GCN | 0.76M | 19.72G | $0.1600 \pm 0.0484$ | $0.1168 \pm 0.0270$ | $0.6926 \pm 0.0808$ | $59.93\% \pm 5.89\%$ | $0.9695 \pm 0.1168$ |
| GraphSAGE | 0.89M | 23.23G | $0.0607 \pm 0.0044$ | $0.0745 \pm 0.0063$ | $0.6651 \pm 0.0458$ | $64.25\% \pm 3.27\%$ | $0.8004 \pm 0.0453$ |
| GAT | 0.76M | 33.02G | $0.2036 \pm 0.0142$ | $0.1040 \pm 0.0130$ | $0.6293 \pm 0.0178$ | $64.94\% \pm 1.87\%$ | $0.9370 \pm 0.0283$ |
| PNA | 2.75M | OOM | - | - | - | - | - |
| DeepGate2 | 1.28M | 24.15G | $0.0406 \pm 0.0004$ | $0.0621 \pm 0.0003$ | $0.6976 \pm 0.0079$ | $63.16\% \pm 0.77\%$ | $0.8003 \pm 0.0083$ |
| DeepGate3 | 8.17M | OOM | - | - | - | - | - |
| PolarGate | 0.88M | 44.48G | $0.7767 \pm 0.3965$ | $0.1179 \pm 0.0615$ | $0.9096 \pm 0.1934$ | $53.00\% \pm 14.82\%$ | $1.8042 \pm 0.3771$ |
| HOGA-2 | 0.78M | 43.12G | $0.1635 \pm 0.0004$ | $0.0896 \pm 0.0002$ | $0.6245 \pm 0.0004$ | $64.81\% \pm 0.42\%$ | $0.8777 \pm 0.0005$ |
| HOGA-5 | 0.78M | OOM | - | - | - | - | - |
| DeepGate4 | 7.37M | 41.09G | $\mathbf{0.0233 \pm 0.0010}$ | $\mathbf{0.0462 \pm 0.0019}$ | $\mathbf{0.4789 \pm 0.0180}$ | $\mathbf{79.00\% \pm 0.30\%}$ | $\mathbf{0.5484 \pm 0.0166}$ |

### A.5 LOGIC EQUIVALENCE CHECKING

Logic Equivalence Checking (LEC) is a critical task in Formal Verification, aimed at determining whether two designs are functionally equivalent. As circuit complexity grows, the significance of LEC increases since design errors in such systems can lead to costly fixes or operational failures in the final product.

We evaluate LEC on the ITC99 dataset by extracting subcircuits with multiple primary inputs (PIs) and a single primary output (PO). Given a subcircuit pair $(G_1, G_2)$, the model performs a binary classification task to predict whether $G_1$ and $G_2$ are equivalent. In the candidate pairs, only 1.29% of pairs are equivalent, highlighting the challenge of imbalanced data. To assess performance, we use the widely adopted metrics Average Precision (AP) and Precision-Recall Area Under the Curve (PR-AUC). These metrics are threshold-independent and particularly effective for imbalanced datasets, where one class is significantly rarer than the other.

Table 10: Logic Equivalence Checking

| Method | AP | PR-AUC |
|--------|-----|--------|
| GCN | 0.05 | 0.04 |
| GraphSAGE | 0.10 | 0.11 |
| GAT | 0.02 | 0.02 |
| PNA | 0.20 | 0.17 |
| HOGA-5 | 0.03 | 0.03 |
| DeepGate2 | 0.13 | 0.13 |
| PolarGate | 0.03 | 0.21 |
| DeepGate3 | OOM | OOM |
| DeepGate3$^\dagger$ | 0.17 | 0.17 |
| DeepGate4 | **0.31** | **0.30** |

Note that DeepGate3$^\dagger$ denotes that we use our proposed updating strategy and training pipline. As shown in Table 10, DeepGate4 outperforms all other methods by a significant margin, achieving the highest AP (0.31) and PR-AUC (0.30), and improve these two metrics by 55% and 42% respectively, compared to the second-best method. These values indicate its superior ability to balance precision and recall, especially in scenarios with imbalanced data.

### A.6 BOOLEAN SATISFIABILITY PROBLEM

The Boolean Satisfiability (SAT) problem is a fundamental computational problem that determines whether a Boolean formula can evaluate to logic-1 for at least one variable assignment. As the first proven NP-complete problem, SAT serves as a cornerstone in computer science, with applications spanning fields such as scheduling, planning, and verification. Modern SAT solvers primarily utilize the conflict-driven clause learning (CDCL) algorithm, which efficiently handles path conflicts during the search process and explores additional constraints to reduce the search space. Over the years, various heuristic strategies have been developed to further accelerate CDCL in SAT solvers.

We follow the setting in DeepGate2 (Shi et al., 2023). We utilize the CaDiCal (QUEUE, 2019) SAT solver as the backbone solver and modify the variable decision heuristic based on it. In the Baseline setting, SAT problems are directly solved using the backbone SAT solver. For model-acclerated SAT solving, given a SAT instance, the first step is to encode the corresponding AIG to get the gate embedding. During the variable decision process, a decision value $d_i$ is assigned to variable $v_i$. If another variable $v_j$ with an assigned value $d_j$ is identified as correlated to $v_i$, the reversed value $d'_j$ is assigned to $v_i$, i.e., $d_i = 0 \ if \ d_j = 1$ or $d_i = 1 \ if \ d_j = 0$. The determination of correlated variables relies on their functional similarity, and the similarity $Sim(v_i, v_j)$ exceeding the threshold $\theta$ indicates correlation.

Table 11: SAT solving time comparsion.† denotes that we use our updating strategy.

| Case | Name | ad44 | f20 | ab18 | ac1 | ad14 | Avg. |
|------|------|------|------|------|------|------|------|
|      | Size | 44949 | 27806 | 37275 | 42038 | 44949 | 39403.4 |
| Baseline | Solving Time | 918.21 | 1046.31 | 3150.81 | 5522.85 | 5766.85 | 3281.01 |
| DeepGate3† | Model Runtime | 27.73 | 16.57 | 22.60 | 33.17 | 27.27 | 25.47 |
|            | Solving Time | 678.42 | 952.91 | 1607.06 | 6189.61 | 4413.96 | 2768.39 |
| PolarGate | Model Runtime | 0.01 | 0.01 | 0.01 | 0.24 | 0.01 | 0.06 |
|           | Solving Time | **606.74** | 1154.87 | **1000.02** | 3923.88 | **3222.98** | 1981.70 |
| Exphormer† | Model Runtime | 0.74 | 0.51 | 0.62 | 0.64 | 0.97 | 0.70 |
|            | Solving Time | 885.98 | 1177.07 | 1293.57 | 4156.04 | 3387.24 | 2179.98 |
| DeepGate4 | Model Runtime | 3.65 | 2.80 | 3.10 | 3.33 | 3.62 | 3.30 |
|           | Solving Time | 970.28 | **143.09** | 1351.49 | **393.25** | 4268.57 | **1425.34** |

The results are shown in Table 11. Since SAT solving is time-consuming, we compare our approach only with the top-3 methods listed in Table 1, namely DeepGate3†, Exphormer†, and PolarGate. The Baseline represents using the SAT solver without any model-based acceleration. Leveraging its exceptional ability to understand the functional relationships within circuits, DeepGate4 achieves a substantial reduction in SAT solving time, with an 86.33% reduction for case *f20* and an 92.90% reduction for case *ac1*. Regarding average solving time, it achieves a 56.56% reduction, outperforming all other methods. These results highlight DeepGate4's strong generalization capability and effectiveness in addressing real-world SAT solving challenges.

