# OpenReview forum: "DeepGate4: Efficient and Effective Representation Learning for Circuit Design at Scale"
_ICLR.cc/2025/Conference — ICLR 2025 Poster_

### Official Review · Reviewer_YSp1 · 2024-11-02

**Soundness:** 3
**Presentation:** 3
**Contribution:** 3
**Rating:** 5
**Confidence:** 4

**Summary:**

The paper presents a novel graph transformer model, DeepGate4, to address the challenges of scalability and efficiency in circuit representation learning, particularly for large-scale circuits used in electronic design automation (EDA) tasks. Previous methods, such as graph neural networks (GNNs) based and transformer-based models, face over-squashing and large memory overhead challenges. DeepGate4 introduces several innovations, including a partitioning method, a GAT-based sparse transformer, and structural encodings, to handle these issues. Experimental results show that DeepGate4 significantly outperforms previous methods.

**Strengths:**

- The proposed method in this paper can reduce GPU memory usage while ensuring the convergence of the model.

**Weaknesses:**

- This paper proposes a method for representation learning but does not provide any experimental validation on specific EDA downstream tasks. Metrics such as the depth of gates provided by the paper are merely custom metrics for representation learning. We hope these methods can be used in practical EDA applications; unfortunately, this paper does not elaborate on how much benefit such methods could bring to actual EDA tasks.
- The novelty of this paper is limited. Previously, Deepgate3$^{[1]}$ also proposed a partitioning strategy for large-scale circuits, which weakens the contribution of the partitioning method.
- Data is very important for deep learning. This paper uses two old datasets, but the data are insufficient for such a large-scale model, and the results provided are not solid enough.

[1] Zhengyuan Shi, Ziyang Zheng, Sadaf Khan, Jianyuan Zhong, Min Li, and Qiang Xu. Deepgate3: Towards scalable circuit representation learning. arXiv:2407.11095.

**Questions:**

- What's the difference between the two partitioning strategies used in Deepgate3 and Deepgate4. It would be better to discuss more about this.
- Conducting the experiments on modern, larger-scale datasets/netlists(e.g., [1], [2]) would significantly enhance the value of the paper.

[1] Chai, Zhuomin, et al. "Circuitnet: An open-source dataset for machine learning in vlsi cad applications with improved domain-specific evaluation metric and learning strategies." IEEE Transactions on Computer-Aided Design of Integrated Circuits and Systems 42.12 (2023): 5034-5047.

[2] https://github.com/TILOS-AI-Institute/MacroPlacement

---

> ### Author Response · Authors · 2024-11-22
>
> We appreciate the reviewer’s suggestion to improve our works. Here are our answers to the weaknesses and questions.
>
> **Q1:** We hope these methods can be used in practical EDA applications; unfortunately, this paper does not elaborate on how much benefit such methods could bring to actual EDA tasks.
>
> **Answer**
> We have add two downstream tasks, Logic Equivalence Checking(LEC) and Boolean Satisfiability Problem(SAT), in Section A.5 and A.6.
>
> **Logic Equivalence Checking(LEC)**
> Logic Equivalence Checking (LEC) is a prominent Formal Verification tasks, which determines whether two given designs are functionally equivalent. As circuit complexity continues to grow, the importance of LEC increases, as design errors in these complex systems can result in expensive fixes or operational failures in the final product.
>
> We evaluate LEC on ITC99 Dataset. We randomly cut subcircuits with multiple PI and one PO from ITC99 evaluation dataset. Given a subciruit pair$(G_1,G_2)$, the model will perform a binary classification task to predict whether $G_1$ and $G_2$ is equivalent or not. In the candidate pairs sequence, only 1.29% of pair is equivalent. We evaluate the performance with widely used metrics Average Precision(AP) and Precision-Recall Area Under the Curve(PR-AUC), which are regardless of threshold and are particularly useful in scenarios with imbalanced datasets (e.g., when one class is much rarer than the other).
>
> |Method|AP|PR-AUC|
> |------|--|------|
> |GCN|0.05|0.04|
> |GraphSAGE|0.10|0.11|
> |GAT|0.02|0.02|
> |PNA|0.20|0.17|
> |HOGA-5|0.03|0.03|
> |DeepGate2|0.13|0.13|
> |PolarGate|0.03|0.21|
> |DeepGate3|OOM|OOM|
> |DeepGate3$^*$|0.17|0.17|
> |DeepGate4|**0.31**|**0.30**|
>
> Note that DeepGate3$^*$ denotes that we use our proposed updating strategy and training pipline. DeepGate4 outperforms all other methods by a significant margin, achieving the highest AP (0.31) and PR-AUC (0.30), and improve these two metrics by 55% and 42% respectively, compared to the second-best method. These values indicate its superior ability to balance precision and recall, especially in scenarios with imbalanced data.

---

> ### Author Response · Authors · 2024-11-22
>
> **Boolean Satisfiability Problem(SAT)**
> The Boolean Satisfiability (SAT) problem is a fundamental computational problem that determines whether a Boolean formula can evaluate to logic-1 for at least one variable assignment. As the first proven NP-complete problem, SAT serves as a cornerstone in computer science, with applications spanning fields such as scheduling, planning, and verification. Modern SAT solvers primarily utilize the conflict-driven clause learning (CDCL) algorithm, which efficiently handles path conflicts during the search process and explores additional constraints to reduce the search space. Over the years, various heuristic strategies have been developed to further accelerate CDCL in SAT solvers.
>
> We follow the setting in DeepGate2[1]. We utilize the CaDiCal[2] SAT solver as the backbone solver and modify the variable decision heuristic based on it. In the Baseline setting, SAT problems are directly solved using the backbone SAT solver. For model-acclerated SAT solving, given a SAT instance, the first step is to encode the corresponding AIG to get the gate embedding. During the variable decision process, a decision value $d_i$ is assigned to variable $v_i$. If another variable $v_j$ with an assigned value $d_j$ is identified as correlated to $v_i$, the reversed value $d_j'$ is assigned to $v_i$, i.e., $d_i = 0\ if\ d_j = 1$ or $d_i = 1\ if\ d_j = 0$. The determination of correlated variables relies on their functional similarity, and the similarity $Sim(v_i, v_j)$ exceeding the threshold $\theta$ indicates correlation.
>
> **Model Runtime**
> ||Case Name|ad44|f20|ab18|ac1|ad14|Avg.|
> |---|---|---|---|---|---|---|---|
> ||Case Size|44949|27806|37275|42038|44949|39403.4|
> |DeepGate3*|ModelRuntime|27.73|16.57|22.60|33.17|27.27|25.47|
> |PolarGate|ModelRuntime|0.01|0.01|0.01|0.24|0.01|0.06|
> |Exphormer*|ModelRuntime|0.74|0.51|0.62|0.64|0.97|0.70|
> |DeepGate4|ModelRuntime|3.65|2.80|3.10|3.33|3.62|3.30|
>
> **SAT Solving Time**
> |Case|Name|ad44|f20|ab18|ac1|ad14|Avg.|
> |---|---|---|---|---|---|---|---|
> |Baseline|SolvingTime|918.21|1046.31|3150.81|5522.85|5766.85|3281.01|
> |DeepGate3*|SolvingTime|678.42|952.91|1607.06|6189.61|4413.96|2768.39|
> |PolarGate|SolvingTime|**606.74**|1154.87|**1000.02**|3923.88|**3222.98**|1981.70|
> |Exphormer*|SolvingTime|885.98|1177.07|1293.57|4156.04|3387.24|2179.98|
> |DeepGate4|SolvingTime|970.28|**143.09**|1351.49|**393.25**|4268.57|**1425.34**|
>
> Since SAT solving is time-consuming, we compare our approach only with the top-3 methods listed in Table 1 in our paper, namely DeepGate3*, Exphormer*, and PolarGate. The Baseline represents using the SAT solver without any model-based acceleration. Leveraging its exceptional ability to understand the functional relationships within circuits, DeepGate4 achieves a substantial reduction in SAT solving time, with an 86.33\% reduction for case *f20* and an 92.90\% reduction for case *ac1*. Regarding average solving time, it achieves a 56.56\% reduction, outperforming all other methods. These results highlight DeepGate4’s strong generalization capability and effectiveness in addressing real-world SAT solving challenges.
>
>
> [1] Z. Shi, H. Pan, S. Khan, M. Li, Y. Liu, J. Huang, H. Zhen, M. Yuan, Z. Chu, and Q. Xu. DeepGate2: Functionality-aware circuit representation learning. 2023 IEEE/ACM ICCAD, IEEE, 2023.
>
> [2]CaDiCaL 2.0, Armin Biere, Tobias Faller, Katalin Fazekas, Mathias Fleury, Nils Froleyks and Florian Pollitt, Proc. Computer Aidded Verification - 26th Intl. Conf. (CAV'24), Lecture Notes in Computer Science (LNCS), vol. 14681, pages 133-152, Springer 2024

---

> ### Author Response · Authors · 2024-11-22
>
> **Q2 and Q3:** The novelty of this paper is limited. Previously, Deepgate3
> also proposed a partitioning strategy for large-scale circuits, which weakens the contribution of the partitioning method. What's the difference between the two partitioning strategies used in Deepgate3 and Deepgate4. It would be better to discuss more about this.
>
> **Answer**
>
> We summarized our contribution as follow:
> * **An updating strategy** tailored for DAGs based on partitioned graph, ensuring that gate embeddings are computed in logical level order, with each gate being processed only once, thus eliminating redundant computations. Note that even with the graph partition, **DeepGate3 is still limited to fine-tuning graphs with up to 50k nodes**, while our proposed updating strategy, which is adaptable to any graph transformer, achieve sub-linear memory complexity and **thus enable efficient training on graphs with millions even billions of nodes**.
> * **A GAT-based sparse transformer** with global virtual edges, reducing both time and memory complexity to linear in a mini-batch. We further introduce structural encodings for transformers on AIGs by incorporating global and local structural encodings in initialized embedding.
> * **An inference acceleration kernel**, Fused-DeepGate4, designed to optimize the inference process of tokenizer and GAT components with well-designed CUDA kernels that fully exploit the unique sparsity patterns of AIGs.
>
> Here we want to strength that the updating strategy is adaptable to any graph transformers. As shown in Tab. 1, GraphGPs, Exphormer, DAGformer and DeepGate3 will suffer from Out-of-Memory when training on large circuits. We further found training GNNs like PNA and HOGA-5 on OpenABC-D[1] on one Nvidia L40 GPU will also encounter OOM error. This denotes that scaling to large circuits still remains a challenge.
> However, our proposed updating strategy enables training these method on large circuits with **sub-linear memory complexity**. Futhermore, the ablation study in Tab.4 has demonstrated the effectiveness of our porposed module.
>
> [1] Chowdhury A B, Tan B, Karri R, et al. Openabc-d: A large-scale dataset for machine learning guided integrated circuit synthesis[J]. arXiv preprint arXiv:2110.11292, 2021.

---

> ### Author Response · Authors · 2024-11-22
>
> **Q4 and Q5:** This paper uses two old datasets, but the data are insufficient for such a large-scale model, and the results provided are not solid enough. Conducting the experiments on modern, larger-scale datasets/netlists(e.g., [2], [3]) would significantly enhance the value of the paper. Conducting the experiments on modern, larger-scale datasets/netlists(e.g., [2], [3]) would significantly enhance the value of the paper.
>
> [2] Chai, Zhuomin, et al. "Circuitnet: An open-source dataset for machine learning in vlsi cad applications with improved domain-specific evaluation metric and learning strategies." IEEE Transactions on Computer-Aided Design of Integrated Circuits and Systems 42.12 (2023): 5034-5047.
>
> [3] https://github.com/TILOS-AI-Institute/MacroPlacement
>
> **Answer**
>
> Currently, our method are designed for AIG netlists, and we haven't started to transfer it to other modalies, like post-mapping netlist[2] and macroplacement[3]. However, we find a recent pubilshed AIG benchmark OpenABC-D[1] with large-scale AIG averaging 91k gates.
>
> We have included a comparison on the OpenABC-D benchmark in Sec. A.4. Details of the dataset statistics are provided in Sec. A.1. Due to the time-consuming process of preparing labels for graph-level tasks, we evaluate only on gate-level tasks. The results are as follows:
>
> |Model|Param.|Mem.|$L_{gate}^{prob}$|$L_{gate}^{tt\_pair}$|$L_{gate}^{con}$|$P^{con}$|$L_{all}$|
> |-----|------|----|----------------|---------------------|----------------|---------|--------|
> |GCN|0.76M|19.72G|0.16±0.048|0.117±0.027|0.693±0.081|59.93%±5.89%|0.970±0.117|
> |GraphSAGE|0.89M|23.23G|0.061±0.004|0.075±0.006|0.665±0.046|64.25%±3.27%|0.800±0.045|
> |GAT|0.76M|33.02G|0.204±0.014|0.104±0.013|0.629±0.018|64.94%±1.87%|0.937±0.028|
> |PNA|2.75M|OOM|-|-|-|-|-|
> |DeepGate2|1.28M|24.15G|0.041±0.000|0.062±0.000|0.698±0.008|63.16%±0.77%|0.800±0.008|
> |DeepGate3|8.17M|OOM|-|-|-|-|-|
> |PolarGate|0.88M|44.48G|0.777±0.397|0.118±0.062|0.910±0.193|53.00%±14.82%|1.804±0.377|
> |HOGA-2|0.78M|43.12G|0.164±0.000|0.090±0.000|0.625±0.000|64.81%±0.42%|0.878±0.001|
> |HOGA-5|0.78M|OOM|-|-|-|-|-|
> |DeepGate4|7.37M|41.09G|**0.023±0.001**|**0.046±0.002**|**0.479±0.018**|**79.00%±0.30%**|**0.548±0.017**|
>
>
> * **Comparison on Effectiveness** DeepGate4 demonstrates outstanding effectiveness across all training tasks. It achieves state-of-the-art performance on all gate-level tasks within the OpenABC-D datasets. Notably, DeepGate4 reduces the overall loss by 31.48\% compared to the second-best method. Moreover, while baseline models struggle with gate connection prediction, DeepGate4 significantly enhances performance in this area, achieving an accuracy of 79\%. This highlights the outstanding ability of DeepGate4 to capture the structural relationships between gates.
> * **Comparison on Efficiency** In terms of efficiency, models like PNA and HOGA-5 encounter out-of-memory (OOM) errors, whereas DeepGate4 can successfully train a graph transformer on large circuits containing over 500k gates.

---

> ### Author Response · Authors · 2024-11-28
>
> Dear Reviewer YSp1,
>
> Thank you for your thoughtful feedback on our paper. We greatly appreciate the time and effort you have invested in reviewing our work.
>
> As today is the final day for paper revisions, we would greatly appreciate it if you could let us know whether our responses have addressed your concerns or if there are any remaining points we can clarify. Your insights would be invaluable in helping us fully address your concerns and refine our work.
>
> We greatly appreciate your consideration. Please feel free to let us know if there are any specific points you would like us to discuss further.
>
> Thank you again for your valuable time and feedback.
>
> Best regards,
>
> Authors

---

> ### Author Response · Authors · 2024-12-02
>
> Dear Reviewer YSp1,
>
> We sincerely appreciate your thoughtful insights and constructive suggestions on our work. As the extended rebuttal deadline approaches, we would be grateful for any further feedback or guidance you may have.
>
> Thank you once again for your time and effort in reviewing our submission. Your expertise and consideration are deeply valued.
>
> Best regards,
>
> Authors

---

### Official Review · Reviewer_Avwf · 2024-11-03

**Soundness:** 2
**Presentation:** 2
**Contribution:** 2
**Rating:** 6
**Confidence:** 3

**Summary:**

This paper introduces DeepGate4, a graph transformer specifically designed for large-scale circuits. Specifically, DeepGate4 proposes a partitioning method and update strategy tailored for circuit graphs, a GATbased sparse transformer optimized for inference, and global and local structural encodings for circuits. Experiments demonstrate the effectiveness of DeepGate4.

**Strengths:**

1.	Circuit representation learning is an interesting and important research topic.
2.	Experiments demonstrate the proposed method significantly outperforms baselines.

**Weaknesses:**

1.	The writing could be improved. (1) It would be more readable if the authors could provide a summary of contributions. (2) The method description is unclear. For example, the graph partition module and the multi-task training objective is unclear.
2.	The novelty of the proposed method is unclear. The authors incorporates many modules into previous circuit representation learning method. However, the novelty and contribution of these modules seems limited.
3.	Experiments are insufficient. (1) The experiments do not compare their method with a recent SOTA method HOGA [1] that is able to scale to large-scale circuits. (2) The evaluation metrics are unclear. The experiments report many metrics, such as $L_{gate}^{prob}$ and $L_{gate}^{con}$. However, a description of these metrics is missing. (3) The experiments do not apply the representation method to downstream tasks, especially real-world industrial tasks. (4) The sensitivity analysis of hyperparameters is missing. Do the authors use the same hyperparameters across all experiments? It would be more convincing if the authors could conduct sensitivity analysis on the hyperparameters $k$ and $\delta$.

[1] Deng, Chenhui, et al. "Less is More: Hop-Wise Graph Attention for Scalable and Generalizable Learning on Circuits." DAC 2024.

**Questions:**

Please see weaknesses for my questions.

---

> ### Author Response · Authors · 2024-11-22
>
> We appreciate the reviewer’s suggestion to improve our works. Here are our answers to the weaknesses and questions.
>
> **Q1 and Q3:** It would be more readable if the authors could provide a summary of contributions. The novelty of the proposed method is unclear. The authors incorporates many modules into previous circuit representation learning method. However, the novelty and contribution of these modules seems limited.
>
> **Answer**
> We have modified Abstract and Introduction, and summarized our contribution as follow:
> * **An updating strategy** tailored for DAGs based on partitioned graph, ensuring that gate embeddings are computed in logical level order, with each gate being processed only once, thus eliminating redundant computations. Note that even with the graph partition, DeepGate3 is still limited to fine-tuning graphs with up to 50k nodes, while our proposed updating strategy, which is adaptable to any graph transformer, achieve sub-linear memory complexity and thus enable efficient training on graphs with millions even billions of nodes.
> * **A GAT-based sparse transformer** with global virtual edges, reducing both time and memory complexity to linear in a mini-batch. We further introduce structural encodings for transformers on AIGs by incorporating global and local structural encodings in initialized embedding.
> * **An inference acceleration kernel**, Fused-DeepGate4, designed to optimize the inference process of tokenizer and GAT components with well-designed CUDA kernels that fully exploit the unique sparsity patterns of AIGs.
>
> Here we want to strength that the updating strategy is adaptable to any graph transformers. As shown in Tab. 1, GraphGPs, Exphormer, DAGformer and DeepGate3 will suffer from Out-of-Memory when training on large circuits. We further found training GNNs like PNA and HOGA-5 on OpenABC-D[1] on one Nvidia L40 GPU will also encounter OOM error. This denotes that scaling to large circuits still remains a challenge.
> However, our proposed updating strategy enables training these method on large circuits with **sub-linear memory complexity**. Futhermore, the ablation study in Tab.4 has demonstrated the effectiveness of our porposed module.
>
> [1] Chowdhury A B, Tan B, Karri R, et al. Openabc-d: A large-scale dataset for machine learning guided integrated circuit synthesis[J]. arXiv preprint arXiv:2110.11292, 2021.
>
> **Q2:** The method description is unclear. For example, the graph partition module and the multi-task training objective is unclear.
>
> **Answer**
> * **Graph Partititon** The graph paritition algorithm is included in Algor.1. We follow the graph partition used in DeepGate3[1], therefore, we don't give a detailed descrption in our paper. To summarize, we sample specific nodes with a stride $\delta$, and cut cones for these nodes with depth $k$.
> Initially, we focus on gathering all the cones, denoted as $cone^k_v$, that terminate at logic level $k$. A cone is defined as:$cone^k_v = \{u \in V : u \preccurlyeq_k v\}$, where $u \preccurlyeq_k v$ signifies that $u$ is a predecessor of $v$ within $k$ logic levels.
> After collecting cones at level $k$, we proceed in strides of $\delta$. Specifically, we gather additional cones whose output gates are situated at level $k + \delta$. This stride-based progression allows us to partition the circuit iteratively, moving step-by-step through its logic levels.
> This iterative process continues until all partitioned areas collectively cover the entire circuit. By systematically partitioning the circuit into cones, we ensure that each section is analyzed within its local logic level context while maintaining global structural coherence.
> * **Training Objective** We follow the setting in DeepGate3[1], and due to the space limitation, we give a detail definition in Sec A.2. These training objectives can be summarized into two types:
>     * **Regression tasks** For regression tasks, like predicting the logic-1 probability, we use L1-Loss as metrics, e.g. $Loss=\|pred-label\|_1$, where $pred$ is the predicted value and $label$ is the true value.
>     * **Classfication tasks** For classfication tasks, like Gate Connection Prediction, we use Binary Cross-Entropy Loss(BCE Loss), e.g. $L_{\text{BCE}} = - \frac{1}{N} \sum_{i=1}^{N} \left[ y_i \log(p_i) + (1 - y_i) \log(1 - p_i) \right]$, where $N$ is the number of samples, $y_i$ is the true label for the $i$-th sample and $p_i$ is the predicted probability for the $i$-th sample.
>
> [1] Z. Shi, Z. Zheng, S. Khan, J. Zhong, M. Li, and Q. Xu. DeepGate3: Towards scalable circuit representation learning. arXiv:2407.11095.

---

> ### Author Response · Authors · 2024-11-22
>
> **Q3:** The experiments do not compare their method with a recent SOTA method HOGA that is able to scale to large-scale circuits.
>
> **Answer**
> We have added HOGA in Tab1 and Tab3. The result demonstrate that DeepGate4 reduce the overall loss by  54.79% on ITC99 and 39.72% on EPFL respectively, compared to HOGA.
>
> **Table1**
> |Model|Param.|Mem.|$L_{gate}^{prob}$|$L_{gate}^{tt\_pair}$|$L_{gate}^{con}$|$P^{con}$|$L_{graph}^{tt}$|$P^{tt}$|$L_{graph}^{tt\_pair}$|$L_{graph}^{ged\_pair}$|$L_{graph}^{size}$|$L_{graph}^{depth}$|$L_{in}$|$P^{in}$|$L_{all}$|
> |-----|------|----|----------------|---------------------|----------------|---------|----------------|--------|----------------------|-----------------------|------------------|-------------------|--------|--------|--------|
> |HOGA-5|0.78M|42.48G|0.204|0.117|0.609|**68.74%**|0.493|0.254|0.1624|0.141|3.56|1.1378|0.571|68.99%|6.99|
> |DeepGate4|7.37M|7.53G|**0.043**|**0.055**|**0.600**|67.22%|**0.315**|**0.136**|**0.0803**|**0.117**|**1.45**|**0.0591**|**0.461**|**79.50%**|**3.16**|
>
>
>
> **Table3**
> |Method|Time(s)|Mem.(MB)|$L_{func}$|$L_{stru}$|$L_{all}$|Time(s)|Mem.(MB)|$L_{func}$|$L_{stru}$|$L_{all}$|
> |------|-------|--------|----------|----------|---------|-------|--------|----------|----------|---------|
> |HOGA-5|0.290|1010|0.98|6.02|6.99|0.648|2006|1.02|6.33|7.35|
> |DeepGate4|2.496|479|**0.49**|**2.68**|**3.16**|2.263|130|**0.79**|**3.64**|**4.43**|
>
>
> We further include a comparison on the OpenABC-D benchmark in Sec. A.4. Details of the dataset statistics are provided in Sec. A.1. Due to the time-consuming process of preparing labels for graph-level tasks, we evaluate only on gate-level tasks. The results are as follows:
>
> |Model|Param.|Mem.|$L_{gate}^{prob}$|$L_{gate}^{tt\_pair}$|$L_{gate}^{con}$|$P^{con}$|$L_{all}$|
> |-----|------|----|----------------|---------------------|----------------|---------|--------|
> |GCN|0.76M|19.72G|0.16±0.048|0.117±0.027|0.693±0.081|59.93%±5.89%|0.970±0.117|
> |GraphSAGE|0.89M|23.23G|0.061±0.004|0.075±0.006|0.665±0.046|64.25%±3.27%|0.800±0.045|
> |GAT|0.76M|33.02G|0.204±0.014|0.104±0.013|0.629±0.018|64.94%±1.87%|0.937±0.028|
> |PNA|2.75M|OOM|-|-|-|-|-|
> |DeepGate2|1.28M|24.15G|0.041±0.000|0.062±0.000|0.698±0.008|63.16%±0.77%|0.800±0.008|
> |DeepGate3|8.17M|OOM|-|-|-|-|-|
> |PolarGate|0.88M|44.48G|0.777±0.397|0.118±0.062|0.910±0.193|53.00%±14.82%|1.804±0.377|
> |HOGA-2|0.78M|43.12G|0.164±0.000|0.090±0.000|0.625±0.000|64.81%±0.42%|0.878±0.001|
> |HOGA-5|0.78M|OOM|-|-|-|-|-|
> |DeepGate4|7.37M|41.09G|**0.023±0.001**|**0.046±0.002**|**0.479±0.018**|**79.00%±0.30%**|**0.548±0.017**|
>
>
> * **Comparison on Effectiveness** DeepGate4 demonstrates outstanding effectiveness across all training tasks. It achieves state-of-the-art performance on all gate-level tasks within the OpenABC-D datasets. Notably, DeepGate4 reduces the overall loss by 31.48\% compared to the second-best method. Moreover, while baseline models struggle with gate connection prediction, DeepGate4 significantly enhances performance in this area, achieving an accuracy of 79\%. This highlights the outstanding ability of DeepGate4 to capture the structural relationships between gates.
> * **Comparison on Efficiency** In terms of efficiency, models like PNA and HOGA-5 encounter out-of-memory (OOM) errors, whereas DeepGate4 can successfully train a graph transformer on large circuits containing over 500k gates.
>
>
> **Q4**: The evaluation metrics are unclear. The experiments report many metrics. However, a description of these metrics is missing.
>
> **Answer**
> In this work, we follow the setting in DeepGate3, and due to the space limitation, we give a detail definition in Sec A.2. These metrics can be summarized into two types:
> * **MAE** For regression tasks, like predicting the logic-1 probability, we use MAE as metrics, e.g. $MAE=\|pred-label\|_1$
> * **Accuracy** For classfication task, like Gate Connection Prediction, we use Accuracy as metrics, e.g. $Acc=\frac{1}{N}\sum_{i=1}^N 1(pred,label)$.

---

> ### Author Response · Authors · 2024-11-22
>
> **Q5:** The experiments do not apply the representation method to downstream tasks
>
> **Answer**
> We have add two downstream tasks, Logic Equivalence Checking(LEC) and Boolean Satisfiability Problem(SAT), in Section A.5 and A.6.
>
> **Logic Equivalence Checking(LEC)**
> Logic Equivalence Checking (LEC) is a prominent Formal Verification tasks, which determines whether two given designs are functionally equivalent. As circuit complexity continues to grow, the importance of LEC increases, as design errors in these complex systems can result in expensive fixes or operational failures in the final product.
>
> We evaluate LEC on ITC99 Dataset. We randomly cut subcircuits with multiple PI and one PO from ITC99 evaluation dataset. Given a subciruit pair$(G_1,G_2)$, the model will perform a binary classification task to predict whether $G_1$ and $G_2$ is equivalent or not. In the candidate pairs sequence, only 1.29% of pair is equivalent. We evaluate the performance with widely used metrics Average Precision(AP) and Precision-Recall Area Under the Curve(PR-AUC), which are regardless of threshold and are particularly useful in scenarios with imbalanced datasets (e.g., when one class is much rarer than the other).
>
> |Method|AP|PR-AUC|
> |------|--|------|
> |GCN|0.05|0.04|
> |GraphSAGE|0.10|0.11|
> |GAT|0.02|0.02|
> |PNA|0.20|0.17|
> |HOGA-5|0.03|0.03|
> |DeepGate2|0.13|0.13|
> |PolarGate|0.03|0.21|
> |DeepGate3|OOM|OOM|
> |DeepGate3$^*$|0.17|0.17|
> |DeepGate4|**0.31**|**0.30**|
>
> Note that DeepGate3$^*$ denotes that we use our proposed updating strategy and training pipeline. DeepGate4 outperforms all other methods by a significant margin, achieving the highest AP (0.31) and PR-AUC (0.30), and improve these two metrics by 55% and 42% respectively, compared to the second-best method. These values indicate its superior ability to balance precision and recall, especially in scenarios with imbalanced data.
>
> **Boolean Satisfiability Problem(SAT)**
> The Boolean Satisfiability (SAT) problem is a fundamental computational problem that determines whether a Boolean formula can evaluate to logic-1 for at least one variable assignment. As the first proven NP-complete problem, SAT serves as a cornerstone in computer science, with applications spanning fields such as scheduling, planning, and verification. Modern SAT solvers primarily utilize the conflict-driven clause learning (CDCL) algorithm, which efficiently handles path conflicts during the search process and explores additional constraints to reduce the search space. Over the years, various heuristic strategies have been developed to further accelerate CDCL in SAT solvers.
>
> We follow the setting in DeepGate2. We utilize the CaDiCal SAT solver as the backbone solver and modify the variable decision heuristic based on it. In the Baseline setting, SAT problems are directly solved using the backbone SAT solver. For model-acclerated SAT solving, given a SAT instance, the first step is to encode the corresponding AIG to get the gate embedding. During the variable decision process, a decision value $d_i$ is assigned to variable $v_i$. If another variable $v_j$ with an assigned value $d_j$ is identified as correlated to $v_i$, the reversed value $d_j'$ is assigned to $v_i$, i.e., $d_i = 0\ if\ d_j = 1$ or $d_i = 1\ if\ d_j = 0$. The determination of correlated variables relies on their functional similarity, and the similarity $Sim(v_i, v_j)$ exceeding the threshold $\theta$ indicates correlation.
>
> **Model Runtime(s)**
> ||Case Name|ad44|f20|ab18|ac1|ad14|Avg.|
> |---|---|---|---|---|---|---|---|
> ||Case Size|44949|27806|37275|42038|44949|39403.4|
> |DeepGate3*|Runtime|27.73|16.57|22.60|33.17|27.27|25.47|
> |PolarGate|Runtime|0.01|0.01|0.01|0.24|0.01|0.06|
> |Exphormer*|Runtime|0.74|0.51|0.62|0.64|0.97|0.70|
> |DeepGate4|Runtime|3.65|2.80|3.10|3.33|3.62|3.30|
>
> **SAT Solving Time(s)**
> |Case|Name|ad44|f20|ab18|ac1|ad14|Avg.|
> |---|---|---|---|---|---|---|---|
> |Baseline|Time|918.21|1046.31|3150.81|5522.85|5766.85|3281.01|
> |DeepGate3*|Time|678.42|952.91|1607.06|6189.61|4413.96|2768.39|
> |PolarGate|Time|**606.74**|1154.87|**1000.02**|3923.88|**3222.98**|1981.70|
> |Exphormer*|Time|885.98|1177.07|1293.57|4156.04|3387.24|2179.98|
> |DeepGate4|Time|970.28|**143.09**|1351.49|**393.25**|4268.57|**1425.34**|
>
> Since SAT solving is time-consuming, we compare our approach only with the top-3 methods listed in Table 1 in our paper, namely DeepGate3*, Exphormer*, and PolarGate. The Baseline means using the SAT solver without any model-based acceleration. Leveraging its exceptional ability to understand the functional relationships within circuits, DeepGate4 achieves a substantial reduction in SAT solving time, with an 86.33\% reduction for case *f20* and an 92.90\% reduction for case *ac1*. Regarding average solving time, it achieves a 56.56\% reduction, outperforming all other methods. These results highlight DeepGate4’s generalization capability and effectiveness in addressing real-world SAT solving problems.

---

> ### Author Response · Authors · 2024-11-22
>
> **Q6:** The sensitivity analysis of hyperparameters is missing. Do the authors use the same hyperparameters across all experiments? It would be more convincing if the authors could conduct sensitivity analysis on the hyperparameters.
>
> **Answer**
> During evaluation, to ensure the fair comparsion, we keep all hyperparameters the same and we compute the mean and std in last 5 epochs to get stable result.
> Furthermore, we provide an ablation study on $k$ and $δ$, illustrating the sensitivity of our model to these hyperparameters:
>
>
>
> |k|$δ$|Train Mem.(GB)|$L_{func}$|$L_{stru}$|$L_{all}$|
> |---|---|-------------|----------|----------|---------|
> |8|8|13|0.46±0.002|2.45±0.06|2.92±0.06|
> |8|6|13|0.49±0.002|2.68±0.07|3.16±0.08|
> |8|4|13|0.47±0.003|2.58±0.10|3.05±0.09|
> |10|8|34|0.46±0.011|3.21±0.07|3.67±0.08|
> |6|4|7|0.46±0.007|2.66±0.06|3.12±0.06|
>
> Here is our observations:
>
>
>
> * Our method is insensitive to $δ$ (note that overlap level is $k - \delta + 1$). Settings such as $(k=8, \delta=8)$, $(k=8, \delta=6)$, and $(k=8, \delta=4)$ demonstrate that our method is not sensitive to overlap ratios, as performance across these settings is similar.
> * Settings such as $(k=8, \delta=6)$, $(k=10, \delta=8)$, and $(k=6, \delta=4)$ maintain the same overlap level but vary in subgraph size. Results demonstrate that increasing $k$ significantly impacts GPU memory usage. Furthermore, larger $k$ will degrade structural task performance. This is because structural tasks rely more heavily on local information, especially for metrics like $L_{graph}^{ged\_pair}$, $L_{graph}^{size}$, and $L_{graph}^{depth}$ (See Section A.2).

---

> > ### Comment · Reviewer_Avwf · 2024-11-25
> >
> > I sincerely appreciate the authors' thorough and well-articulated responses, which have addressed most of my major concerns. Thus, I will  raise my score to 5. However, one concern remains regarding the details of the proposed updating strategy. The proposed updating strategy may represent an interesting and novel contribution to the community. However, it is still unclear how the proposed updating strategy achieves sub-linear memory complexity and how it differs from existing scalable GNN methods.

---

> > > ### Author Response · Authors · 2024-11-25
> > >
> > > Thank you for recognizing the novelty of our updating strategy. Here are our answers regarding the model complexity.
> > >
> > > **Q1:** how the proposed updating strategy achieves sub-linear memory complexity?
> > >
> > >
> > > **A1:** Our updating strategy achieves sub-linear memory complexity primarily by leveraging historical embedding, as explained in Section 3.4.
> > >
> > > - **Memory Complexity of General GNNs and Graph Transformers**
> > >   General GNNs and Graph Transformers encode all the nodes in a graph. This means that **the embeddings of all nodes are processed on the GPU** during both the forward and backward passes. Consequently, the memory complexity is:
> > >   - $O(n)$ for GNNs and Sparse Graph Transformers.
> > >   - $O(n^2)$ for Dense Graph Transformers.
> > >
> > > - **Our Proposed Updating Strategy**
> > >   Our strategy fully exploits circuit properties based on the observations in Section 3.3. Specifically, for given PIs, the output of a gate **is determined only by the fan-in cone** of that gate. Therefore, encoding all nodes simultaneously is unnecessary. Instead, we partitioned circuits into cones, group cones into mini-batches, and encode the mini-batches iteratively following the logic levels. After encoding a mini-batch, we save their embedding **on CPU instead of GPU** with the corrsponding index for future use.
> > >
> > >   This denotes that, with our method, **only the embeddings of nodes within a mini-batch are processed on the GPU**. Since a cone has a maximum size ($2^k-1$), and with a fixed mini-batch size (e.g., 128 in our paper), there exists an **upper bound on memory usage**, which is independent of the circuit size. This guarantees sub-linear memory complexity of our method.
> > >
> > > - **Additional Optimizations:**
> > >   Other proposed modules further reduce memory usage, as highlighted in Table 4:
> > >   - Using marks to avoid overlapping computations.
> > >   - Implementing a GAT-based sparse attention mechanism with global virtual edges.
> > >   - Employing an inference acceleration CUDA kernel.
> > >
> > >
> > > **Q2:** how it differs from existing scalable GNN methods？
> > >
> > > **A2:**
> > >
> > > We compare our method with HOGA and general scalable GNNs, highlighting our advantages in **efficiency** and **effectiveness**.
> > >
> > > **Compare with HOGA**
> > >
> > > In the original paper of HOGA, the memory complexity and memory usage were not explicitly discussed. HOGA's main idea is to pool all hop nodes into a hop embedding, which is then aggregated for each gate using Gated Self-Attention. Consequently, the memory complexity of HOGA is estimated to be $O(n)$.
> > >
> > > Experiments on the OpenABC-D benchmark further confirm this, showing that when trained on large circuits, HOGA-5 encounters out-of-memory (OOM) errors and HOGA-2 requires more GPU memory than general GNNs such as GCN and GraphSAGE, despite having a similar number of parameters. This highlights the **memory inefficiency** of HOGA.
> > >
> > > **Compare with General Scalable GNNs**
> > >
> > > The differences between our approach and general scalable GNNs are elaborated in Section 2. To summarize, applying general scalable GNNs to AIGs remains challenging because these models often disregard the causal relationships between subgraphs by using completely random sampling techniques. In contrast, when modeling circuit functionality as a computational graph, it is critical to adhere to strict topological constraints to preserve the hierarchical and causal dependencies inherent in the circuit design. Random sampling methods commonly employed in general GNNs fail to capture these essential properties, resulting in suboptimal performance for circuit-specific tasks.
> > >
> > > Moreover, experimental results on the ITC99, EPFL, and OpenABC-D benchmarks demonstrate that general GNNs lack adaptability for circuit-related applications, leading to **ineffectiveness**.

---

> ### Author Response · Authors · 2024-11-28
>
> Dear Reviewer Avwf,
>
> Thank you for your thoughtful feedback on our paper. We greatly appreciate the time and effort you have invested in reviewing our work.
>
> As today is the final day for paper revisions, we would greatly appreciate it if you could let us know whether our responses have addressed your concerns or if there are any remaining points we can clarify. Your insights would be invaluable in helping us fully address your concerns and refine our work.
>
> We greatly appreciate your consideration. Please feel free to let us know if there are any specific points you would like us to discuss further.
>
> Thank you again for your valuable time and feedback.
>
> Best regards,
>
> Authors

---

> > ### Comment · Reviewer_Avwf · 2024-11-30
> >
> > I sincerely appreciate the authors' responses, which have addressed my remaining concerns. The proposed updating strategy well incorporates the circuit domain knowledge into the model design. Thus, I have raised my score accordingly. I strongly encourage the authors to incorporate these clarifications into the revised manuscript to further enhance the work.

---

### Official Review · Reviewer_ArxE · 2024-11-04

**Soundness:** 3
**Presentation:** 2
**Contribution:** 3
**Rating:** 8
**Confidence:** 4

**Summary:**

This paper proposes a scalable and efficient graph transformer specifically designed for large-scale circuits, namely DeepGate4. DeepGate4 consists of several key components: (1) a partition method and update strategy tailored for circuit graphs; (2) a GAT-based sparse transformer optimized for inference by leveraging the sparse nature of circuits; (3) global and local structural encodings for circuits, along with a loss balancer. Experimental results show that DeepGate4 significantly outperforms previous state-of-the-art (SOTA) method.

**Strengths:**

This paper addresses a fundamental problem in machine learning for electronic design automation. The proposed DeepGate4 consists of several interesting and effective modules to improve the scalability of the method. Experimental results demonstrate a significant improvement.

**Weaknesses:**

The novelty of the proposed graph partitioning method and the loss balancer is limited. The graph partition mechanism is a commonly-used technique for handling large-scale industrial circuits. Moreover, the dynamic loss balancer has been widely investigated in computer vision [1].

The OpenABC-D benchmark is widely-used in previous work. Can the authors evaluate their method on this benchmark as well. A recent SOTA method called HOGA [2] also addresses the challenge of scaling to large-scale circuits. Can the authors compare their method with HOGA.

[1] Lin, Tsung-Yi, et al. "Focal Loss for Dense Object Detection." Proceedings of the IEEE International Conference on Computer Vision. 2017.
[2] Deng, Chenhui, et al. "Less is More: Hop-Wise Graph Attention for Scalable and Generalizable Learning on Circuits." DAC 2024.

**Questions:**

1. Can the authors explain the metrics used in Table 1?
2. Can the authors clarify whether they evaluate their method on unseen testing datasets or training datasets?
3. Can the proposed method be applied to sequential circuit representation learning?

---

> ### Author Response · Authors · 2024-11-20
>
> We appreciate the reviewer’s suggestion to improve our works. Here are our answers to the weaknesses and questions.
>
> **Q1:** Explain the metrics
>
> **Answer**
> In this work, we follow the setting in DeepGate3[1], and due to the space limitation, we give a detail definition in Sec A.2. These metrics can be summarized into two types:
> * **MAE** For regression tasks, like predicting the logic-1 probability, we use MAE as metrics, e.g. $MAE=\|pred-label\|_1$
> * **Accuracy** For classfication task, like Gate Connection Prediction, we use Accuracy as metrics, e.g. $Acc=\frac{1}{N}\sum_{i=1}^N 1(pred,label)$.
>
> [1] Z. Shi, Z. Zheng, S. Khan, J. Zhong, M. Li, and Q. Xu. DeepGate3: Towards scalable circuit representation learning. arXiv:2407.11095.
>
> **Q2:** Clarify the evaluation
>
> **Answer**
> All models are evaluated on unseen testing datasets. To ensure a fair comparison, we use identical hyperparameters across all models and compute the mean and standard deviation over the last five epochs to obtain stable results.
>
> **Q3:** Apply to sequential circuit?
>
> **Answer**
>
> Currently, we evaluate our method on combinational circuits. However, since the proposed updating strategy is adaptable with any graph transformer on DAGs, we believe it has the potential to be extended to sequential circuits.
>
> **Q4:** Limitd novelty.
>
> **Answer**
> We have modified Abstract and Introduction, and summarized our contribution as follow:
> * **An updating strategy** tailored for DAGs based on partitioned graph, ensuring that gate embeddings are computed in logical level order, with each gate being processed only once, thus eliminating redundant computations. Note that even with the graph partition, DeepGate3 is still limited to fine-tuning graphs with up to 50k nodes, while our proposed **intra-level and inter-level updating strategy**, which is adaptable to any graph transformer as shown in Tab.1, achieve sub-linear memory complexity and thus enable efficient training on graphs with millions even billions of nodes.
> * **A GAT-based sparse transformer** with global virtual edges, reducing both time and memory complexity to linear in a mini-batch. We further introduce structural encodings for transformers on AIGs by incorporating global and local structural encodings in initialized embedding.
> * **An inference acceleration kernel**, Fused-DeepGate4, designed to optimize the inference process of tokenizer and GAT components with well-designed CUDA kernels that fully exploit the unique sparsity patterns of AIGs.

---

> > ### Author Response · Authors · 2024-11-20
> >
> > **Q5:** Experiment on OpenABC-D.
> >
> > **Answer**
> > We have included a comparison on the OpenABC-D benchmark in Sec. A.4. Details of the dataset statistics are provided in Sec. A.1. Due to the time-consuming process of preparing labels for graph-level tasks, we evaluate only on gate-level tasks. The results are as follows:
> >
> > |Model|Param.|Mem.|$L_{gate}^{prob}$|$L_{gate}^{tt\_pair}$|$L_{gate}^{con}$|$P^{con}$|$L_{all}$|
> > |-----|------|----|----------------|---------------------|----------------|---------|--------|
> > |GCN|0.76M|19.72G|0.16±0.048|0.117±0.027|0.693±0.081|59.93%±5.89%|0.970±0.117|
> > |GraphSAGE|0.89M|23.23G|0.061±0.004|0.075±0.006|0.665±0.046|64.25%±3.27%|0.800±0.045|
> > |GAT|0.76M|33.02G|0.204±0.014|0.104±0.013|0.629±0.018|64.94%±1.87%|0.937±0.028|
> > |PNA|2.75M|OOM|-|-|-|-|-|
> > |DeepGate2|1.28M|24.15G|0.041±0.000|0.062±0.000|0.698±0.008|63.16%±0.77%|0.800±0.008|
> > |DeepGate3|8.17M|OOM|-|-|-|-|-|
> > |PolarGate|0.88M|44.48G|0.777±0.397|0.118±0.062|0.910±0.193|53.00%±14.82%|1.804±0.377|
> > |HOGA-2|0.78M|43.12G|0.164±0.000|0.090±0.000|0.625±0.000|64.81%±0.42%|0.878±0.001|
> > |HOGA-5|0.78M|OOM|-|-|-|-|-|
> > |DeepGate4|7.37M|41.09G|**0.023±0.001**|**0.046±0.002**|**0.479±0.018**|**79.00%±0.30%**|**0.548±0.017**|
> >
> >
> > * **Comparison on Effectiveness** DeepGate4 demonstrates outstanding effectiveness across all training tasks. It achieves state-of-the-art performance on all gate-level tasks within the OpenABC-D datasets. Notably, DeepGate4 reduces the overall loss by 31.48\% compared to the second-best method. Moreover, while baseline models struggle with gate connection prediction, DeepGate4 significantly enhances performance in this area, achieving an accuracy of 79\%. This highlights the outstanding ability of DeepGate4 to capture the structural relationships between gates.
> > * **Comparison on Efficiency** In terms of efficiency, models like PNA and HOGA-5 encounter out-of-memory (OOM) errors, whereas DeepGate4 can successfully train a graph transformer on large circuits containing over 500k gates.
> >
> > **Q6:** Compare with HOGA:
> >
> > **Answer**
> > We have added HOGA in Tab1 and Tab3. The result demonstrate that DeepGate4 reduce the overall loss by  54.79% on ITC99 and 39.72% on EPFL respectively, compared to HOGA.
> >
> > **Table1**
> > |Model|Param.|Mem.|$L_{gate}^{prob}$|$L_{gate}^{tt\_pair}$|$L_{gate}^{con}$|$P^{con}$|$L_{graph}^{tt}$|$P^{tt}$|$L_{graph}^{tt\_pair}$|$L_{graph}^{ged\_pair}$|$L_{graph}^{size}$|$L_{graph}^{depth}$|$L_{in}$|$P^{in}$|$L_{all}$|
> > |-----|------|----|----------------|---------------------|----------------|---------|----------------|--------|----------------------|-----------------------|------------------|-------------------|--------|--------|--------|
> > |HOGA-5|0.78M|42.48G|0.204|0.117|0.609|**68.74%**|0.493|0.254|0.1624|0.141|3.56|1.1378|0.571|68.99%|6.99|
> > |DeepGate4|7.37M|7.53G|**0.043**|**0.055**|**0.600**|67.22%|**0.315**|**0.136**|**0.0803**|**0.117**|**1.45**|**0.0591**|**0.461**|**79.50%**|**3.16**|
> >
> >
> > **Table3**
> > |Method|Time(s)|Mem.(MB)|$L_{func}$|$L_{stru}$|$L_{all}$|Time(s)|Mem.(MB)|$L_{func}$|$L_{stru}$|$L_{all}$|
> > |------|-------|--------|----------|----------|---------|-------|--------|----------|----------|---------|
> > |HOGA-5|0.290|1010|0.98|6.02|6.99|0.648|2006|1.02|6.33|7.35|
> > |DeepGate4|2.496|479|**0.49**|**2.68**|**3.16**|2.263|130|**0.79**|**3.64**|**4.43**|

---

> > > ### Comment · Reviewer_ArxE · 2024-11-26
> > >
> > > Thanks for the authors' effort. I appreciate the additional experiments on OpenABC-D and the comparison with HOGA, which makes the  performance of the proposed method more clear. The detailed response addresses my concerns and I raise my score accordingly.

---

### Official Review · Reviewer_vLL8 · 2024-11-04

**Soundness:** 3
**Presentation:** 4
**Contribution:** 3
**Rating:** 6
**Confidence:** 4

**Summary:**

This work aims to address the challenges in scaling to large circuits due to limitations like over-squashing in graph neural networks and the quadratic complexity of transformer-based models by introducing DeepGate4, a scalable and efficient graph transformer specifically designed for large-scale circuits. DeepGate4 has following contributions: (1) design a partitioning method and update strategy tailored for circuit graphs, reducing memory complexity to sub-linear levels; (2) propose a GAT based sparse transformer optimized for inference by leveraging the sparse nature of circuits; (3) propose with a loss balancer that dynamically adjusts the weights of multitask losses to stabilize training.

**Strengths:**

1. The experimental evaluation is comprehensive and compelling, showing significant performance improvements over state-of-the-art baselines. The empirical results strongly validate the effectiveness of the proposed approach, with substantial gains across multiple metrics and circuit scales.
2. The technical contribution is substantial. The work demonstrates innovation in addressing fundamental scalability challenges in circuit analysis through:
  a. A novel graph partitioning strategy
  b. An efficient sparse transformer architecture
  c. An adaptive loss balancing mechanism
3. The paper presents a thorough and well-structured literature review. It effectively traces the evolution of GNN applications in EDA while critically analyzing the limitations of current transformer-based approaches in GNN architectures. This comprehensive background provides a clear motivation for the proposed methodologies and firmly establishes their theoretical foundations.

**Weaknesses:**

1. The results of the experiment can be represented more visually in graphs. It will be better if the authors could explain in detail the parameter settings in the ablation experiment. For example, why was the parameter k set to 8?
2. It is mentioned that the loss balancer is very effective in reducing the loss, can the authors expand more on this or give a theoretical analysis? And it will be better if the authors could analyze why the loss balancer must be introduced in the last layer.
3、The description in section 3.3 is not very clear. It will be better if the authors could elaborate on the role of receptive fields and the specific way and basis for defining them in inter-class and intra-class.
4、How does the setting of k and d have effects on computing the receptive fields and subsequent calculations for the cones?
5、The figures need to be further explained clearly. For example, what are the meanings of the red arrows and the black arrows in Figure 2?

**Questions:**

Please see weakness

---

> ### Author Response · Authors · 2024-11-17
>
> We thank the reviewer for their valuable feedback. Here are our answers to the weaknesses and questions.
>
> **Q1 and Q4**:How does the setting of k and d have effects the experiment?
>
> **Answer**
> We have added an ablation study on $k$ and $\delta$ in  Section A.3 in our paper. These parameters influence memory usage and overlap levels as follows:
> - **$k$ (Maximum Level)**: Determines subgraph size, with size always smaller than $2^{k+1}-1$. Larger $k$ increases GPU memory usage significantly.
> - **$\delta$ (Stride)**: Defines overlap regions, with overlap level as $k - \delta + 1$.
>
> Furthermore, we provide an ablation study on $k$ and $δ$, illustrating the sensitivity of our model to these hyperparameters:
>
> | k   | $δ$ | Train Mem.(GB) | $L_{func}$          | $L_{stru}$          | $L_{all}$          |
> | --- | -------- | ----------- | ------------------- | ------------------- | ------------------- |
> | 8   | 8        | 13    | 0.46 $\pm$ 0.002 | 2.45 $\pm$ 0.06 | 2.92 $\pm$ 0.06 |
> | 8   | 6        | 13    | 0.49 $\pm$ 0.002 | 2.68 $\pm$ 0.07 | 3.16 $\pm$ 0.08 |
> | 8   | 4        | 13    | 0.47 $\pm$ 0.003 | 2.58 $\pm$ 0.10 | 3.05 $\pm$ 0.09 |
> | 10   | 8        | 34    | 0.46 $\pm$ 0.011 | 3.21 $\pm$ 0.07 | 3.67 $\pm$ 0.08 |
> | 6   | 4        | 7    | 0.46 $\pm$ 0.007 | 2.66 $\pm$ 0.06 | 3.12 $\pm$ 0.06 |
>
> Here is our observations:
>
> * Our method is insensitive to $δ$, as shown by similar performance across $(k=8, δ=8)$, $(k=8, δ=6)$, and $(k=8, δ=4)$.
> * Results on $(k=8, δ=6)$, $(k=10, δ=8)$, and $(k=6, δ=4)$ demonstrate that increasing $k$ impacts GPU memory usage. Furthermore, larger $k$ will degrade structural task since they rely more heavily on local information, especially for metrics like $L_{graph}^{ged\_pair}$ and $L_{graph}^{size}$ (See Sec A.2).
>
> **Q2**: Can the authors expand more on loss balancer?
>
> **Answer**
> The analysis of loss balancer based on gradient norms has been widely studied in various domains[1, 2]. We briefly include an analysis with a simple example:
> * **Gradient-Based Balancing**: Assume there are three losses $L_1$, $L_2$, and $L_3$, and the total loss is $L = L_1 + L_2 + L_3$. During updating, the gradient for model parameter $w$ at iteration $t$ is: $w_{t+1}=w_t-\frac{\partial L_1}{\partial w_t}-\frac{\partial L_2}{\partial w_t}-\frac{\partial L_3}{\partial w_t}$
> Here, $\frac{\partial L_i}{\partial w_t}$ represents the contribution of each loss. Since loss scales may significantly due to the difference definition, the loss balancer adjusts weights to equalize gradient contributions:$w_{t+1}=w_t-a_1\frac{\partial L_1}{\partial w_t}-a_2\frac{\partial L_2}{\partial w_t}-a_3\frac{\partial L_3}{\partial w_t}$
> where $a_i = \frac{1}{\|\frac{\partial L_i}{\partial w_t}\|_2}$. This ensures the l2-norm of $a_i\frac{\partial L_i}{\partial w_t}$ is the same to balanced updates for each loss term, independent of their scale.
> * **Why Last-Layer**:
> (1)It ensures the influence of the entire encoder is considered. Balancing earlier layers may ignore gradient contributions from subsequent layers.
> (2)It is computationally efficient, as balancing earlier layers requires computing gradients for all encoder layers, increasing memory usage and time consumption.
>
> [1]Chen Z. et al., "Gradnorm: Gradient normalization for adaptive loss balancing in deep multitask networks," ICML, 2018, pp. 794–803.
> [2]Défossez A. et al., "High fidelity neural audio compression," arXiv preprint, arXiv:2210.13438, 2022.
>
> **Q3**: The description in section 3.3 is not very clear.
>
> **Answer**
> We have modified the Sec 3.3 in our paper for better clarity. We define the receptive field for a node $v$ in a subgraph $cone$ as $\{u\in cone: u\preccurlyeq_k v\}$. In other words, it includes all nodes in the fan-in region of $v$ within the subgraph $cone$.
>
> * **Intra-level overalp**: As shown in Fig 3(a), the receptive field of node $v$ in the overlap region is identical for both the blue and orange cones, as they share the same fan-in region.
> * **Inter-level overlap**: For node $v$ in the overlap region, the receptive field of the blue cone is the fan-in region within the **blue cone**, as shown in Fig 3(b), while the receptive field of oragen cone is the fan-in region within the **orange cone**, as shown in Fig 3\(c). The receptive field shown in Fig 3\(c) can be regarded as the grean area in Fig 3(b) constrained by the orange cone.
>
> The receptive field affects the computations of the GNN tokenizer and sparse transformer, as they aggregate embeddings within the receptive field for node $v$.
>
> **Q5**:The figures need to be further explained clearly.
>
> **Answer**: We have add the caption of the arrows in our paper.
> * **Fig 1**: The black arrows denote edges in the AIG graph, while red arrows indicate message aggregation of DG2.
> * **Fig 2**: The orange arrows denote that we move the node embedding from GPU to CPU, and the blue arrows denote that we move the node embedding from CPU to GPU.

---

### Author Response · Authors · 2024-11-25

Dear Reviewers,

Thank you for your thoughtful feedback on our paper. We greatly appreciate the time and effort you have invested in reviewing our work. During the rebuttal phase, we have provided detailed responses to the concerns you raised, particularly regarding **contribution** and **experiments**, which we believe are central to convey the core ideas and impact of our paper.

If you have any additional thoughts or if there are remaining points of ambiguity, we would be grateful for your further guidance or clarification. Your insights would be invaluable in helping us fully address your concerns and refine our work.

We greatly appreciate your consideration. Please feel free to let us know if there are any specific points you would like us to discuss further.

Thank you once again for your valuable time and feedback.


Best regards,

Authors

---

### Meta-Review · Area_Chair_pmer · 2024-12-21

**Metareview:**

**Summary:** The paper introduces DeepGate4, a graph transformer tailored for large-scale circuit representation learning with the following innovations: (1) a sub-linear memory partitioning and update strategy, (2) a GAT-based sparse transformer optimized for circuit sparsity and supported by dedicated CUDA kernels. The model achieves non-trivial improvements over state-of-the-art methods and demonstrates practical utility across various EDA tasks.

**Strength:**

1. Scalability and efficiency: The sub-linear memory approach and CUDA optimizations allow DeepGate4 to handle circuits with millions of nodes, addressing the efficiency concern of prior work.

2. Comprehensive experiments: Strong empirical results across multiple downstream EDA tasks, establish the model’s effectiveness.

**Weakness:**

1. Limited novelty: Some contributions, like the partitioning strategy and loss balancing, are incremental extensions of prior work.
2. Evaluation on more EDA tasks: The evaluation on older datasets limits the broader applicability of the results; experiments on more complex EDA tasks would strengthen the paper.
3. Lack of clarity in presentation: Certain technical details (e.g., graph partitioning, parameter settings) and figures could be clearer to improve reproducibility and readability.

**Reasons for the decision:**

While novelty and dataset limitations exist, the paper’s strong empirical results and practical impact outweigh these concerns. As such, I’m inclined to accept this paper. Revisions should address the clarity issues and add evaluations on more datasets to further strengthen the work.

**Additional Comments On Reviewer Discussion:**

During the rebuttal period, several key points were raised by reviewers, which were answered in detail by the authors:

**1. Evaluation on downstream tasks**

Reviewers’ concern (YSp1): Lack of practical EDA task evaluations.

Author response: Added Logic Equivalence Checking (LEC) and SAT tasks, demonstrating non-trivial performance gains.

**2. Novelty and comparisons with necessary baselines**

Reviewers’ concern (ArxE and Avwf): Incremental contributions over DeepGate3 and lack of comparison with HOGA.

Author response: Clarified scalability improvements and added HOGA comparisons, showing non-trivial gains.

**3. Clarity and presentation**

Reviewers’ concern (vLL8 and Avwf): Unclear method descriptions and insufficient figure captions.

Author response: Revised text and added detailed figure explanations.

Overall, the rebuttal effectively addressed major concerns, particularly regarding downstream task evaluations and HOGA comparisons. Although dataset limitations and incremental novelty persist, the paper's practical contributions, scalability, and strong empirical results justify its acceptance.

---

### Decision · Program_Chairs · 2025-01-22

Accept (Poster)